# What Can We Learn From The Selective Prediction And Uncertainty Estimation Performance Of 523 Imagenet Classifiers?

**Ido Galil**
Technion
idogalil.ig@gmail.com

**Mohammed Dabbah**
Amazon
m.m.dabbah@gmail.com

**Ran El-Yaniv**
Technion, Deci.AI
rani@cs.technion.ac.il

## ABSTRACT

When deployed for risk-sensitive tasks, deep neural networks must include an uncertainty estimation mechanism. Here we examine the relationship between deep architectures and their respective training regimes, with their corresponding selective prediction and uncertainty estimation performance. We consider some of the most popular estimation performance metrics previously proposed including AUROC, ECE, AURC as well as coverage for selective accuracy constraint. We present a novel and comprehensive study of selective prediction and the uncertainty estimation performance of 523 existing pretrained deep ImageNet classifiers that are available in popular repositories. We identify numerous and previously unknown factors that affect uncertainty estimation and examine the relationships between the different metrics. We find that distillation-based training regimes consistently yield better uncertainty estimations than other training schemes such as vanilla training, pretraining on a larger dataset and adversarial training. Moreover, we find a subset of ViT models that outperform any other models in terms of uncertainty estimation performance. For example, we discovered an unprecedented 99% top-1 selective accuracy on ImageNet at 47% coverage (and 95% top-1 accuracy at 80%) for a ViT model, whereas a competing EfficientNet-V2-XL cannot obtain these accuracy constraints at any level of coverage. Our companion paper, also published in ICLR 2023 (Galil et al., 2023), examines the performance of these classifiers in a class-out-of-distribution setting.

## 1 INTRODUCTION

The excellent performance of deep neural networks (DNNs) has been demonstrated in a range of applications, including computer vision, natural language understanding and audio processing. To deploy these models successfully, it is imperative that they provide an uncertainty quantification of their predictions, either via some kind of *selective prediction* or a probabilistic confidence score.

Notwithstanding, what metric should we use to evaluate the uncertainty estimation performance? There are many and diverse ways so the answer to this question is not obvious, and to demonstrate the difficulty, consider the case of two classification models for the stock market that predict whether a stock's value is about to increase, decrease, or remain neutral (three-class classification). Suppose that model A has a 95% true accuracy, and generates a confidence score of 0.95 on every prediction (even on misclassified instances); model B has a 40% true accuracy, but always gives a confidence score of 0.6 on correct predictions, and 0.4 on incorrect ones. Model B can be utilized easily to generate perfect investment decisions. Using *selective prediction* (El-Yaniv & Wiener, 2010; Geifman & El-Yaniv, 2017), Model B will simply reject all investments on stocks whenever the confidence score is 0.4. While model A offers many more investment opportunities, each of its predictions carries a 5% risk of failure.

Among the various metrics proposed for evaluating the performance of uncertainty estimation are: *Area Under the Receiver Operating Characteristic* (AUROC or AUC), *Area Under the Risk-Coverage curve* (AURC) (Geifman et al., 2018), selective risk or coverage for a *selective accuracy constraint* (SAC), *Negative Log-likelihood* (NLL), *Expected Calibration Error* (ECE), which is often used for evaluating a model's *calibration* (see Section 2) and *Brier score* (Brier, 1950). All these metrics

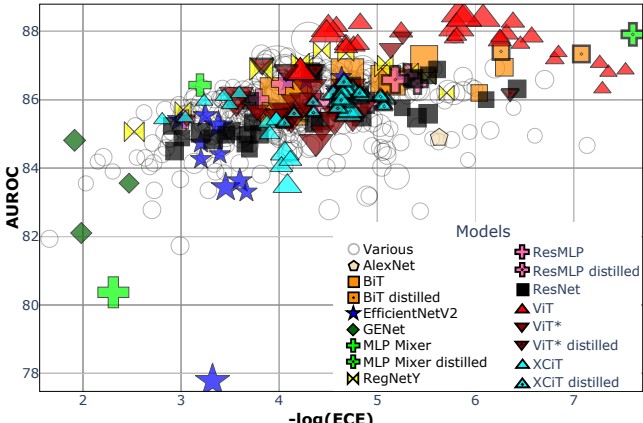

Figure 1: A comparison of 523 models by their AUROC ($\times 100$, higher is better) and -log(ECE) (higher is better) on ImageNet. Each marker's size is determined by the model's number of parameters. A full version graph is given in Figure 8. Distilled models are better than non-distilled ones. A subset of ViT models is superior to all other models for all aspects of uncertainty estimation ("ViT" in the legend, marked as a red triangle facing upwards); the performance of EfficientNet-V2 and GENet models is worse.

are well known and are often used for comparing the uncertainty estimation performance of models (Moon et al., 2020; Nado et al., 2021; Maddox et al., 2019; Lakshminarayanan et al., 2017). Somewhat surprisingly, NLL, Brier, AURC, and ECE all fail to reveal the uncertainty superiority of Model B in our investment example (see Appendix A for the calculations). Both AUROC and SAC, on the other hand, reveal the advantage of Model B perfectly (see Appendix A for details). It is not hard to construct counterexamples where these two metrics fails and others (e.g., ECE) succeed. To sum up this brief discussion, we believe that the ultimate suitability of a performance metric should be determined by its context. If there is no specific application in mind, there is a strong incentive to examine a variety of metrics, as we choose to do in this study.

This study evaluates the ability of 523 models from the Torchvision and Timm repositories (Paszke et al., 2019; Wightman, 2019) to estimate uncertainty[1]. Our study identifies several major factors that affect confidence rankings, calibration, and selective prediction, and lead to **numerous empirical contributions** important to selective predictions and uncertainty estimation. While no new algorithm or method is introduced in our paper, our study generates many interesting conclusions that will help practitioners achieve more powerful uncertainty estimation. Moreover, the research questions that are uncovered by our empirical study shed light on uncertainty estimation, which may stimulate the development of new methods and techniques for improving uncertainty estimation. Among the most interesting conclusions our study elicits are:

(1) **Knowledge distillation training improves estimation**. Training regimes incorporating any kind of *knowledge distillation* (KD) (Hinton et al., 2015) lead to DNNs with improved uncertainty estimation performance evaluated by any metric, more than by using any other training tricks (such as pretraining on a larger dataset, adversarial training, etc.). In Galil et al. (2023) we find similar performance boosts for *class-out-of-distribution* (C-OOD) detection.

(2) **Certain architectures are more inclined to perform better or worse at uncertainty estimation**. Some architectures seem more inclined to perform well on all aspects of uncertainty estimation, e.g., a subset of vision transformers (ViTs) (Dosovitskiy et al., 2021) and the zero-shot language–vision CLIP model (Radford et al., 2021), while other architectures tend to perform worse, e.g., EfficientNet-V2 and GENet (Tan & Le, 2021; Lin et al., 2020). These results are visualized in Figure 1. In Galil et al. (2023) we find that ViTs and CLIPs are also powerful C-OOD detectors.

(3) **Several training regimes result in a subset of ViTs that outperforms all other architectures and training regimes.** These regimes include the original one from the paper introducing ViTs (Dosovitskiy et al., 2021; Steiner et al., 2022; Chen et al., 2022; Ridnik et al., 2021). These ViTs

---

[1]Our code is available at https://github.com/IdoGalil/benchmarking-uncertainty-estimation-performance

achieve the best uncertainty estimation performance on any aspect measured, both in absolute terms and per-model size (# parameters, see Figures 9 and 10 in Appendix B).

(4) **Temperature scaling improves selective and ranking performance**. The simple post-training calibration method of *temperature scaling* (Guo et al., 2017), which is known to improve ECE, for the most part also improves ranking (AUROC) and selective prediction—meaning not only does it calibrate the probabilistic estimation for each individual instance, but it also improves the partial order of all instances induced by those improved estimations, pushing instances more likely to be correct to have a higher confidence score than instances less likely to be correct (see Section 3).

(5) **The correlations between AUROC, ECE, accuracy and the number of parameters are dependent on the architecture analyzed**. Contrary to previous work by (Guo et al., 2017), we observe that while there is a strong correlation between accuracy/number of parameters and ECE or AUROC within each specific family of models of the same architecture, the correlation flips between a strong negative and a strong positive correlation depending on the type of architecture being observed. For example, as DLA (Yu et al., 2018) architectures increase in size and accuracy, their ECE deteriorates while their AUROC improves. The exact opposite, however, can be observed in XCiTs (Ali et al., 2021) as their ECE improves with size while their AUROC deteriorates (see Appendix L).

(6) **The best model in terms of AUROC or SAC is not always the best in terms of calibration**, as illustrated in Figure 1, and the trade-off should be considered when choosing a model based on its application.

## 2 HOW TO EVALUATE DEEP UNCERTAINTY ESTIMATION PERFORMANCE

Let $\mathcal{X}$ be the input space and $\mathcal{Y}$ be the label space. Let $P(\mathcal{X}, \mathcal{Y})$ be an unknown distribution over $\mathcal{X} \times \mathcal{Y}$. A model $f$ is a prediction function $f : \mathcal{X} \to \mathcal{Y}$, and its predicted label for an image $x$ is denoted by $\hat{y}_f(x)$. The model's *true* risk w.r.t. $P$ is $R(f|P) = E_{P(\mathcal{X}, \mathcal{Y})}[\ell(f(x), y)]$, where $\ell : \mathcal{Y} \times \mathcal{Y} \to \mathbb{R}^+$ is a given loss function, for example, 0/1 loss for classification. Given a labeled set $S_m = \{(x_i, y_i)\}_{i=1}^m \subseteq (\mathcal{X} \times \mathcal{Y})$, sampled i.i.d. from $P(\mathcal{X}, \mathcal{Y})$, the *empirical risk* of model $f$ is $\hat{r}(f|S_m) \triangleq \frac{1}{m} \sum_{i=1}^m \ell(f(x_i), y_i)$. Following Geifman et al. (2018), for a given model $f$ we define a *confidence score* function $\kappa(x, \hat{y}|f)$, where $x \in \mathcal{X}$, and $\hat{y} \in \mathcal{Y}$ is the model's prediction for $x$, as follows. The function $\kappa$ should quantify confidence in the prediction of $\hat{y}$ for the input $x$, based on signals from model $f$. This function should induce a partial order over instances in $\mathcal{X}$.

The most common and well-known $\kappa$ function for a classification model $f$ (with softmax at its last layer) is its softmax response values: $\kappa(x, \hat{y}|f) \triangleq f(x)_{\hat{y}}$ (Cordella et al., 1995; De Stefano et al., 2000). We chose to focus on studying uncertainty estimation performance using softmax response as the models' $\kappa$ function because of its extreme popularity, and its importance as a baseline due to its solid performance compared to other methods (Geifman & El-Yaniv, 2017; Geifman et al., 2018). While this is the main $\kappa$ we evaluate, we also test the popular uncertainty estimation technique of *Monte Carlo dropout* (MC dropout) (Gal & Ghahramani, 2016), which is motivated by Bayesian reasoning. Although these methods use the direct output from $f$, $\kappa$ could be a different model unrelated to $f$ and unable to affect $f$'s predictions. Note that to enable a probabilistic interpretation, $\kappa$ can only be calibrated if its values reside in $[0, 1]$ whereas for ranking and selective prediction any value in $\mathbb{R}$ can be used.

A *selective model* $f$ (El-Yaniv & Wiener, 2010; Chow, 1957) uses a *selection function* $g : \mathcal{X} \to \{0, 1\}$ to serve as a binary selector for $f$, enabling it to abstain from giving predictions for certain inputs. $g$ can be defined by a threshold $\theta$ on the values of a $\kappa$ function such that $g_\theta(x|\kappa, f) = \mathbb{1}[\kappa(x, \hat{y}_f(x)|f) > \theta]$. The performance of a selective model is measured using coverage and risk, where *coverage*, defined as $\phi(f, g) = E_P[g(x)]$, is the probability mass of the non-rejected instances in $\mathcal{X}$. The *selective risk* of the selective model $(f, g)$ is defined as $R(f, g) \triangleq \frac{E_P[\ell(f(x), y)g(x)]}{\phi(f, g)}$. These quantities can be evaluated empirically over a finite labeled set $S_m$, with the *empirical coverage* defined as $\hat{\phi}(f, g|S_m) = \frac{1}{m} \sum_{i=1}^m g(x_i)$, and the *empirical selective risk* defined as $\hat{r}(f, g|S_m) \triangleq \frac{\frac{1}{m} \sum_{i=1}^m \ell(f(x_i), y_i)g(x_i)}{\hat{\phi}(f, g|S_m)}$. Similarly, SAC is defined as the largest coverage available for a specific accuracy constraint. A way to visually inspect the behavior of a $\kappa$ function for selective prediction can be done using the risk-coverage (RC) curve (El-Yaniv & Wiener, 2010)—a curve showing the selective risk as a function of coverage, measured on some chosen test set; see Figure 2 for an

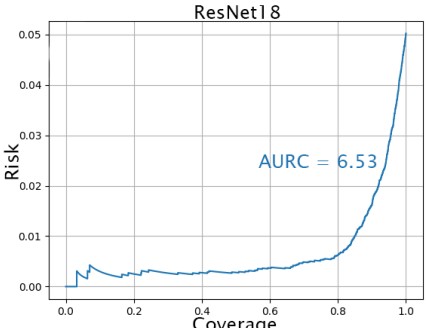

Figure 2: The RC curve made by a ResNet18 trained on CIFAR-10, measured on the test set. The risk is calculated using a 0/1 loss (meaning the model has about 95% accuracy for 1.0 coverage); the $\kappa$ used was softmax-response. The value of the risk at each point of coverage corresponds to the selective risk of the model when rejecting inputs that are not covered at that coverage slice. e.g., the selective risk for coverage 0.8 is about 0.5%, meaning that an end user setting a matching threshold would enjoy a model accuracy of 99.5% on the 80% of images the model would not reject.

example. In general, though, two RC curves are not necessarily comparable if one does not fully dominate the other (Figure 3 shows an example of lack of dominance).

The AURC and E-AURC metrics were defined by (Geifman et al., 2018) for quantifying the selective quality of $\kappa$ functions via a single number, with AURC being defined as the area under the RC curve. AURC, however, is very sensitive to the model's accuracy, and in an attempt to mitigate this, E-AURC was suggested. The latter also suffers from sensitivity to accuracy, as we demonstrate in Appendix C. The advantage of scalar metrics such as the above is that they summarize the model's overall uncertainty estimation behavior by reducing it to a single scalar. When not carefully chosen, however, these reductions could result in a loss of vital information about the problem (recall the investment example from Section 1, which is also discussed in Appendix A: reducing an RC curve to an AURC does not show that Model B has an optimal 0 risk if the coverage is smaller than 0.4). Thus, the choice of the "correct" single scalar performance metric unfortunately must be task-specific. When comparing the uncertainty estimation performance of deep architectures that exhibit different accuracies, we find that AUROC and SAC can effectively "normalize" accuracy differences that plague the usefulness of other metrics (see Figure 3). This normalization is essential in our study where we compare uncertainty performance of hundreds of models that can greatly differ in their accuracies.

For risk-sensitive deployment, let us consider the two models in Figure 3 ; EfficientNet-V2-XL (Tan & Le, 2021) and ViT-B/32-SAM (Chen et al., 2022). While the former model has better overall accuracy and AURC (metrics that could lead us to believe the model is best for our needs), it cannot guarantee a Top-1 ImageNet selective accuracy above 95% for any coverage. ViT-B/32-SAM, on the other hand, can provide accuracies above 95% for all coverages below 50%.

In applications where risk (or coverage) constraints are dictated (Geifman & El-Yaniv, 2017), the most straightforward and natural metric is SAC (or selective risk), which directly measures the coverage (resp., risk) given at the required level of risk (resp., coverage) constraint. We demonstrate this in Appendix I, evaluating which models give the most coverage for an ambitious SAC of 99%. If instead a specific range of coverages is specified, we could measure the area under the RC curve for those coverages: $\text{AURC}_{\mathcal{C}}(\kappa, f|S_m) = \frac{1}{|\mathcal{C}|} \sum_{c \in \mathcal{C}} \hat{r}(f, g_c|S_m)$, with $\mathcal{C}$ being those required coverages.

Often, these requirements are not known or can change as a result of changing circumstances or individual needs. Also, using metrics sensitive to accuracy such as AURC makes designing architectures and methods to improve $\kappa$ very hard, since an improvement in these metrics could be attributed to either an increase in overall accuracy (if such occurred) or to a real improvement in the model's ranking performance. Lastly, some tasks might not allow the model to abstain from making predictions at all, but instead require interpretable and well-calibrated probabilities of correctness, which could be measured using ECE.

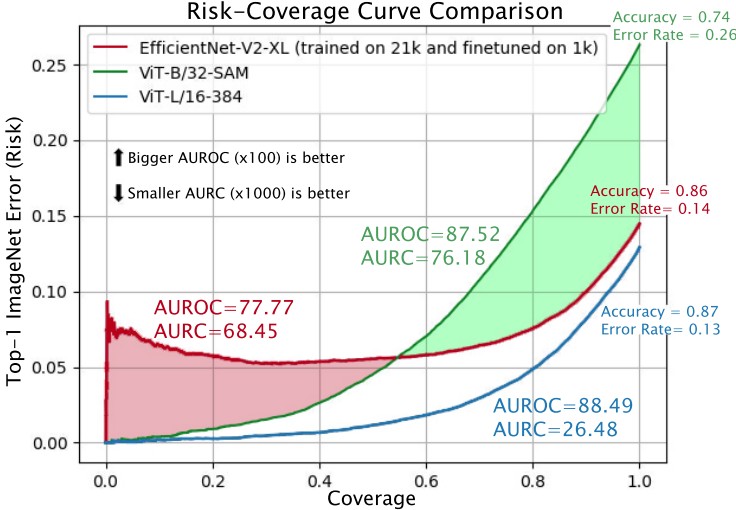

Figure 3: A comparison of RC curves made by the best (ViT-L/16-384) and worst (EfficientNet-V2-XL) models we evaluated in terms of AUROC. Comparing ViT-B/32-SAM to EfficientNet-V2 exemplifies the fact that neither accuracy nor AURC reflect selective performance well enough.

## 2.1 MEASURING RANKING AND CALIBRATION

A $\kappa$ function is not necessarily able to change the model's predictions. Therefore, it can improve the selective risk by ranking correct and incorrect predictions better, inducing a more accurate partial order over instances in $\mathcal{X}$. Thus, for every two random samples $(x_1, y_1), (x_2, y_2) \sim P(\mathcal{X}, \mathcal{Y})$ and given that $\ell(f(x_1), y_1) > \ell(f(x_2), y_2)$, the *ranking* performance of $\kappa$ is defined as the probability that $\kappa$ ranks $x_2$ higher than $x_1$:

$$\mathbf{Pr}[\kappa(x_1, \hat{y}|f) < \kappa(x_2, \hat{y}|f) | \ell(f(x_1), y_1) > \ell(f(x_2), y_2)] \tag{1}$$

We discuss this definition in greater detail in Appendix D. The AUROC metric is often used in the field of machine learning. When the 0/1 loss is in play, it is known that AUROC in fact equals the probability in Equation (1) (Fawcett, 2006) and thus is a proper metric to measure ranking in classification (AKA discrimination). AUROC is furthermore equivalent to Goodman and Kruskal's $\gamma$-correlation (Goodman & Kruskal, 1954), which for decades has been extensively used to measure ranking (known as "resolution") in the field of metacognition (Nelson, 1984). The precise relationship between $\gamma$-correlation and AUROC is $\gamma = 2 \cdot \text{AUROC} - 1$ (Higham & Higham, 2018). We note also that both the $\gamma$-correlation and AUROC are nearly identical or closely related to various other correlations and metrics; $\gamma$-correlation (AUROC) becomes identical to Kendall's $\tau$ (up to a linear transformation) in the absence of tied values. Both metrics are also closely related to *rank-biserial correlation*, the *Gini coefficient* (not to be confused with the measure from economics) and the *Mann–Whitney U test*, hinting at their importance and usefulness in a variety of fields and settings. In Appendix E, we briefly compare the ranking performance of deep neural networks and humans in metacognitive research, and in Appendix F we address criticism of using AUROC to measure ranking.

The most widely used metric for calibration is ECE (Naeini et al., 2015). For a finite test set of size $N$, ECE is calculated by grouping all instances into $m$ interval bins (such that $m \ll N$), each of size $\frac{1}{m}$ (the confidence interval of bin $B_j$ is $(\frac{j-1}{m}, \frac{j}{m}]$). With acc($B_j$) being the mean accuracy in bin $B_j$ and conf($B_j$) being its mean confidence, ECE is defined as

$$ECE = \sum_{j=1}^{m} \frac{|B_j|}{N} \sum_{i \in B_j} \left| \frac{\mathbb{1}[\hat{y}_f(x_i) = y_i]}{|B_j|} - \frac{\kappa(x, \hat{y}_f(x_i)|f)}{|B_j|} \right|$$

$$= \sum_{j=1}^{m} \frac{|B_j|}{N} |\text{acc}(B_j) - \text{conf}(B_j)|$$

Since ECE is widely accepted we use it here to evaluate calibration, and follow (Guo et al., 2017) in setting the number of bins to $m = 15$. Many alternatives to ECE exist, allowing an adaptive binning scheme, evaluating the calibration on the non-chosen labels as well, and other various methods (Nixon et al., 2019; Vaicenavicius et al., 2019; Zhao et al., 2020). Relevant to our objective is that by using binning, this metric is not affected by the overall accuracy as is the Brier score (mentioned in Section 1), for example.

## 3 PERFORMANCE ANALYSIS

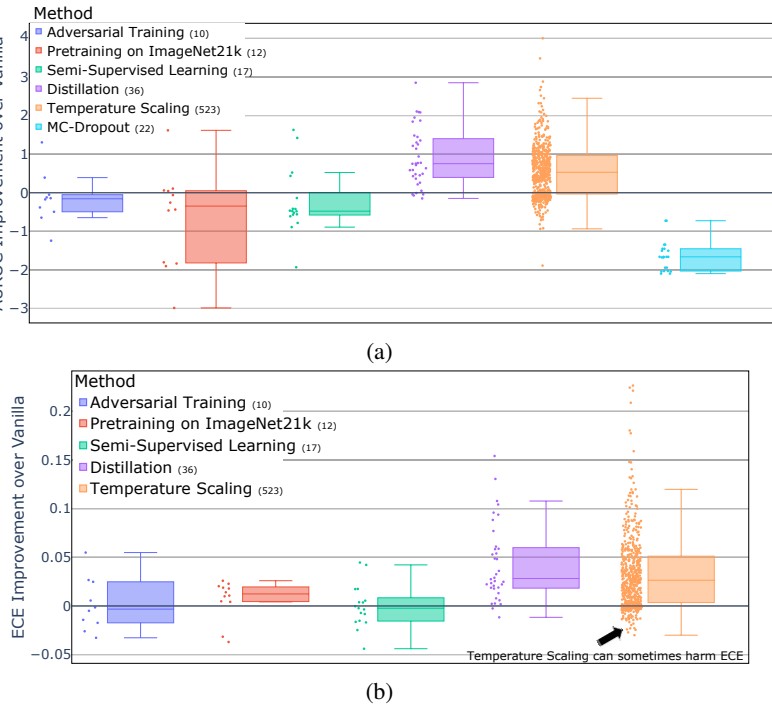

Figure 4: A comparison of different methods and their improvement in terms of (a) AUROC and (b) ECE, relative to the same model's performance without employing the method. Markers above the x-axis represent models that benefited from the evaluated method, and vice versa. The numbers in the legend to the right of each method indicate the number of pairs compared. Temperature scaling can sometimes harm ECE, even though its purpose is to improve it.

In this section we study the performance of 523 different models (available in timm 0.4.12 and torchvision 0.10). Note that all figures from our analysis are available as interactive plotly plots in the supplementary material, which provides information about every data point.

1) **Among the training regimes evaluated, knowledge distillation improves performance the most**. We evaluated several training regimes: (a) Training that involves KD in any form, including Touvron et al. (2021b), knapsack pruning with distillation (in which the teacher is the original unpruned model) (Aflalo et al., 2020) and a pretraining technique that employs distillation (Ridnik et al., 2021); (b) adversarial training (Xie et al., 2020a; Tramèr et al., 2018); (c) pretraining on ImageNet21k ("pure", with no additions) (Tan & Le, 2021; Touvron et al., 2021a; 2022); and (d) various forms of weakly or semi-supervised learning (Mahajan et al., 2018; Yalniz et al., 2019; Xie et al., 2020b). To make a fair comparison, we only compare pairs of models such that both models have identical architectures and training regimes, with the exception of the method itself being evaluated (e.g., training with or without knowledge distillation). More information about each data point of comparison is available in the supplementary material. Note that the samples are of various sizes due to the different number of potential models available for each.

Of the methods mentioned above, training methods incorporating distillation improve AUROC and ECE the most. For example, looking at Figure 4a, it is evident that distillation (purple box) almost

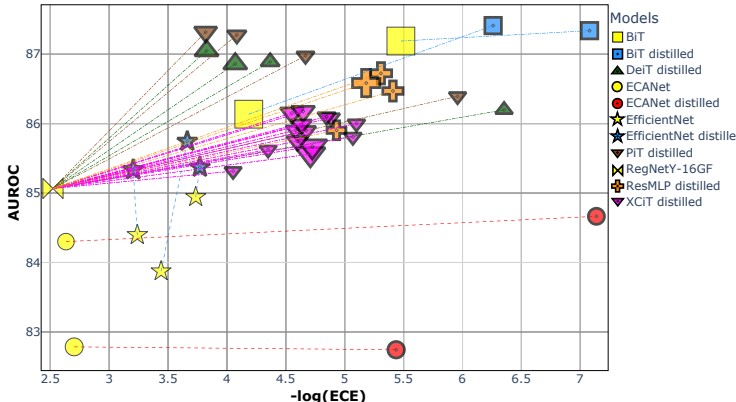

Figure 5: Comparing teacher models (yellow markers) to their KD students (represented by markers with thick borders and a dot). The performance of each model is measured in AUROC (higher is better) and -log(ECE) (higher is better).

always improves AUROC, and moreover, its median improvement is the best of all techniques evaluated. The same observation can be made with regards to improving ECE; see Figure 4b. Distillation seems to greatly improve both metrics even when the teacher itself is much worse at both metrics. Figure 5 nicely shows this by comparing the teacher architecture and the students in each case. Additionally, in a pruning scenario that included distillation in which the original model was also the teacher (Aflalo et al., 2020), the pruned models outperformed their teachers. The fact that KD improves the model over its original form is surprising, and suggests that the distillation process itself helps uncertainty estimation. In Galil et al. (2023) we find that KD also improves C-OOD detection performance, measured by AUROC. We discuss these effects in greater detail in Appendix G.

2) **Temperature scaling greatly benefits AUROC and selective prediction**. Evaluations of the simple post-training calibration method of temperature scaling (TS) (Guo et al., 2017), which is widely known to improve ECE without changing the model's accuracy, also revealed several interesting facts: (a) TS consistently and greatly improves AUROC and selective performance (see Figure 4a)—meaning not only does TS calibrate the probabilistic estimation for each individual instance, but it also improves the partial order of all instances induced by those improved estimations. While TS is well known and used for calibration, to the best of our knowledge, its benefits for selective prediction were previously *unknown*. (b) While TS is usually beneficial, it could harm some models (see Figures 4a and 4b). While it is surprising that TS—a calibration method—would harm ECE, this phenomenon is explained by the fact that TS optimizes NLL and not ECE (to avoid trivial solutions), and the two may sometimes misalign. (c) Models that benefit from TS in terms

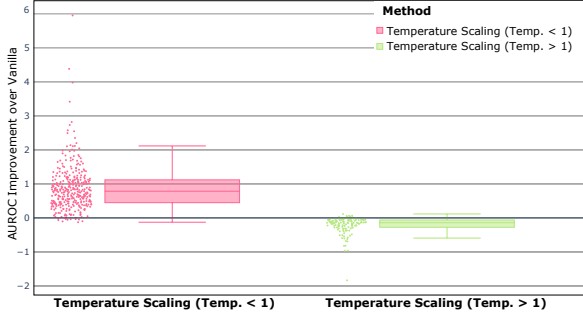

Figure 6: Out of 523 models evaluated, models that were assigned a temperature higher than 1 by the calibration process tended to degrade in AUROC performance rather than improve. Markers above the x-axis represent models that benefited from TS, and vice versa.

of AUROC tend to have been assigned a temperature smaller than 1 by the calibration process (see Figure 6). This, however, does not hold true for ECE (see Figure 14 in Appendix H). This example

also emphasizes the fact that improvements in terms of AUROC do not necessarily translate into improvements in ECE, and vice versa. (d) While all models usually improve with TS, the overall ranking of uncertainty performance between families tends to stay similar, with the worse (in terms of ECE and AUROC) models closing most of the gap between them and the mediocre ones (see Figure 13 in Appendix H). .

3) **A subset of ViTs outperforms all other architectures in selective prediction, ranking and calibration, both in absolute terms and per-model size**. Several training regimes (including the original regime from the paper introducing ViT) Dosovitskiy et al. (2021); Steiner et al. (2022); Chen et al. (2022); Ridnik et al. (2021) result in ViTs that outperform all other architectures and training regimes in terms of AUROC and ECE (see Figure 1; Figure 13 in Appendix H shows this is true even after using TS) as well as for the SAC of 99% we explored (see Figure 7 and Appendix I). These ViTs also outperform all other models in terms of C-OOD detection performance (Galil et al., 2023). Moreover, for any size, ViT models outperform their competition in all of these metrics (see Figures 9 and 10 in Appendix B and Figure 15 in Appendix I).

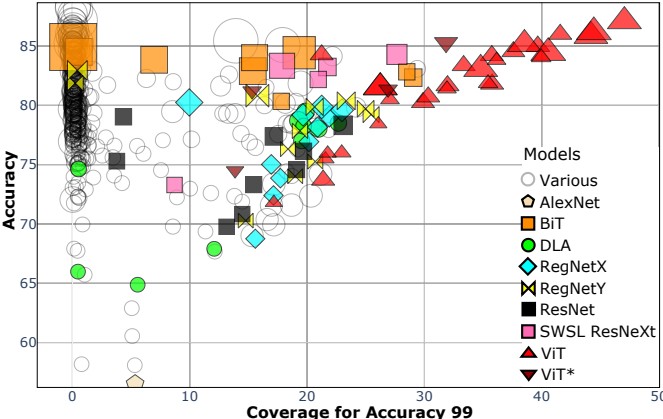

Figure 7: Comparison of models by their overall accuracy and the coverage they are able to provide given a selective accuracy constraint of Top-1 99% on ImageNet. A higher coverage is better. Only ViT models are able to provide coverage beyond 30% for this constraint. They provide more coverage than any other model compared to their accuracy or size. "Various" refers to all other models (out of the 523) that were not mentioned by name.

Further research into other training regimes, however, reveals that not all training regimes result in superb performance (Touvron et al., 2021b; 2022; Singh et al., 2022; Paszke et al., 2019) (these ViTs are dubbed "ViT*" in the figures), even when a similar amount of data is introduced into the training and strong augmentations are used. In fact, the models trained by Chen et al. (2022) were not pretrained at all and yet reach superb ranking. Even the largest model introduced by Tran et al. (2022), which is a large modified ViT that was pretrained on JFT-4B (a dataset containing 4 billion images) with the aim of excelling in uncertainty estimation (and other areas), is outperformed by the best ViT we evaluated: Plex L achieves an AUROC of 87.7 (while its smaller versions, Plex M and Plex S, achieve an AUROC of 87.4 and 86.7, respectively), compared to 88.5 achieved by ViT-L/16-384 that has less parameters and was pretrained on ImageNet-21k. In total, 18 ViTs trained on ImageNet-21k outperform[2] Plex L, among which are two variations of small ViTs (each with 36 or 22 million parameters). In Appendix J we analyze the different hyperparameters and augmentations used for training the ViT models evaluated in this paper. Unfortunately, no clear conclusions emerge to explain the advantage of the successful training regimes. There is, however, ample evidence to show that advanced augmentations are unlikely to be part of such an explanation.

The above facts suggest that the excellent performance exhibited by some ViTs cannot be attributed to the amount of data or to the augmentations used during training. These observations warrant

---

[2] The authors had not released clear results for Plex ECE performance on ImageNet, making a comparison of calibration difficult. The authors mentioned that the average ECE of Plex L in CIFAR-10, CIFAR-100 and ImageNet is slightly below 0.01. Our evaluations revealed six ViTs that achieved the same results, with the most calibrated model being ViT-T/16 with an ECE of 0.0054 on ImageNet.

additional research with the hope of either training more robust ViTs or transferring the unidentified ingredient of the successful subset of ViTs into other models.

4) **Correlations between AUROC, ECE, accuracy and the model's size could either be positive or negative, and depend on the family of architectures evaluated. This observation contradicts previous smaller scale studies on calibration.** While AUROC and ECE are (negatively) correlated (they have a Spearman correlation of -0.44, meaning that generally, as AUROC improves, so does ECE), their agreement on the best performing model depends greatly on the architectural family in question. For example, the Spearman correlation between the two metrics evaluated on 28 undistilled XCiTs is 0.76 (meaning ECE deteriorates as AUROC improves), while for the 33 ResNets (He et al., 2016) evaluated, the correlation is -0.74. Another general observation is that contrary to previous work by (Guo et al., 2017) concerning ECE, the correlations between ECE and the accuracy or the number of model parameters are nearly *zero*, although each family tends to have a strong correlation, either negative or positive. We include a family-based comparison in Appendix L for correlations between AUROC/ECE and accuracy, number of parameters and input size. These results suggest that while some architectures might utilize extra resources to achieve improved uncertainty estimation capabilities, other architectures do not and are even harmed in this respect.

5) **The zero-shot language–vision CLIP model is well-calibrated, with its best instance outperforming 96% of all other models**. CLIP (Radford et al., 2021) enables zero-shot classification and demonstrates impressive performance. We find it is also inclined to be well-calibrated. See Appendix K for details about how we use CLIP. The most calibrated CLIP is based on ViT-B/32 with a linear-probe added to it, and obtains an ECE of 1.3%, which outperforms 96% of models evaluated. Moreover, for their size category, CLIP models tend to outperform their competition in calibration, with the exception of ViTs (see Figure 10 in Appendix B). While this trend is clear for zero-shot CLIPs, we note that some models' calibration performance deteriorates with the addition of a linear-probe. Further research is required to understand the ingredients of multimodal models' contribution to calibration, and to find ways to utilize them to get better calibrated models. For example, could a multimodal pretraining regime be used to get better calibrated models?

6) **MC dropout does not improve selective performance, in accordance with previous works**. We evaluate the performance of MC dropout using predictive entropy as its confidence score and 30 dropout-enabled forward passes. We do not measure its effects on ECE since entropy scores do not reside in $[0, 1]$. Using MC dropout causes a consistent drop in both AUROC and selective performance compared with using the same models with softmax as the $\kappa$ (see Appendix M and Figure 4a). MC dropout's underperformance was also previously observed in (Geifman & El-Yaniv, 2017). We note, however, that evaluations we have conducted in Galil et al. (2023) show that MC dropout performs well when dealing with C-OOD data.

## 4 CONCLUDING REMARKS

We presented a comprehensive study of the effectiveness of numerous DNN architectures (families) in providing reliable uncertainty estimation, including the impact of various techniques on improving such capabilities. Our study led to many new insights and perhaps the most important ones are: (1) architectures trained with distillation almost always improve their uncertainty estimation performance, (2) temperature scaling is very useful not only for calibration but also for ranking and selective prediction, and (3) no DNN (evaluated in this study) demonstrated an uncertainty estimation performance comparable—in any metric tested—to a subset of ViT models (see Section 3).

Our work leaves open many interesting avenues for future research and we would like to mention a few. Perhaps the most interesting question is why distillation is so beneficial in boosting uncertainty estimation. Next, is there an architectural secret in vision transformers (ViT) that enables their uncertainty estimation supremacy under certain training regimes? This issue is especially puzzling given the fact that comparable performance is not observed in many other supposedly similar transformer-based models that we tested. If the secret is not in the architecture, what is the mysterious ingredient of the subset of training regimes that produces such superb results, and how can it be used to train other models? Finally, can we create specialized training regimes (e.g., Geifman & El-Yaniv (2019)), specialized augmentations, special pretraining regimes (such as CLIP's multimodal training regime) or even specialized neural architecture search (NAS) strategies that can promote superior uncertainty estimation performance?

## ACKNOWLEDGMENTS

This research was partially supported by the Israel Science Foundation, grant No. 710/18.

We thank Prof. Rakefet Ackerman for her help with understanding how uncertainty estimation performance is evaluated for humans in the field of metacognition, and for her useful comments for Appendix E.

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

## A    THE INVESTMENT EXAMPLE

Let us consider two classification models for the stock market that predict whether a stock's value is about to increase, decrease or remain neutral (three-class classification). Suppose that Model A has a 95% true accuracy, and generates a confidence score of 0.95 on any prediction (even on misclassified instances); Model B has a 40% true accuracy, but always gives a confidence score of 0.6 on correct predictions, and 0.4 on incorrect ones. We now try and evaluate these two models using the uncertainty metrics mentioned in Section 1 to see which can reveal Model B's superior uncertainty estimation performance. AURC will fail due to its sensitivity to accuracy (the AURC of Model B is 0.12, more than twice as bad as the AURC for Model A, which is 0.05). NLL will rank Model A four times higher (Model A's NLL is 0.23 and Model B's is 0.93). The Brier score would also much prefer Model A (giving it a score of 0.096 while giving Model B a score of 0.54). Evaluating the models' calibration with ECE will also not reveal Model B's advantages, since it is less calibrated than Model A, which has perfect calibration (Model A has an ECE of 0, and Model B has a worse ECE of 0.4).

AUROC, on the other hand, would give Model B a perfect score of 1 and a terrible score of 0.5 to Model A. The selective risk for Model B would be better for any *coverage* of stock predictions below 40%, and for any SAC above 95% the coverage for Model A would be 0, but 0.4 for Model B.

Those two metrics are not perfect for any example. Let us instead compare two different models for the task of predicting the weather when we cannot abstain from making predictions. Accordingly, being required to provide an accurate probabilistic uncertainty estimation of the model's predictions, AUROC and selective risk would be meaningless (due to the model's inability to abstain in this task), but ECE or the Brier Score would better evaluate the performance the new task requires.

## B    RANKING AND CALIBRATION VISUAL COMPARISON

A comparison of 523 models by their AUROC ($\times 100$, higher is better) and -log(ECE) (higher is better) on ImageNet is visualized in Figure 8. An interactive version of this figure is provided as supplementary material. To compare models fairly by their size, we plot two graphs with the logarithm of the number of parameters as the X-axis, so that models sharing the same x value can be compared solely based on their y value. In Figure 9 we set the X axis to be AUROC (higher is better), and see ViTs outperform any other architecture with a comparable amount of parameters by a large margin. We can also observe that using distillation creates a consistent improvement in AUROC. In Figure 10 we set the X axis to be the negative logarithm of ECE (higher is better) and observe a very similar trend, with ViT outperforming its competition for any model size.

## C    DEMONSTRATION OF E-AURC'S DEPENDENCE ON THE MODEL'S ACCURACY

*Excess-AURC* (E-AURC) was suggested by Geifman et al. (2018) as an alternative to AURC (explained in Section 2). To calculate E-AURC, two AURC scores need to be calculated: (1) $AURC(model)$, the AURC value of the actual model and (2) $AURC(model^*)$, the AURC value of a hypothetical model with identical predicted labels as the first model, but that outputs confidence values that induce a perfect partial order on the instances in terms of their correctness. The latter means that all incorrectly predicted instances are assigned confidence values lower than the correctly predicted instances.

E-AURC is then defined as $AURC(model) - AURC(model^*)$. In essence, this metric acknowledges that given a model's accuracy, the area of $AURC(model^*)$ is always unavoidable no matter how good the partial order is, but anything above that could have been minimized if the $\kappa$ function was better at ranking, assigning correct instances higher values than incorrect ones and inducing a better partial order over the instances.

This metric indeed helps to reduce some of the sensitivity to accuracy suffered by AURC, and for the example presented in Section 1, E-AURC would have given a perfect score of 0 to the model inducing a perfect partial order by its confidence values (Model B). It is easy, however, to craft examples showing that E-AURC prefers models with higher accuracy, even if they have lower or equal capacity to rank.

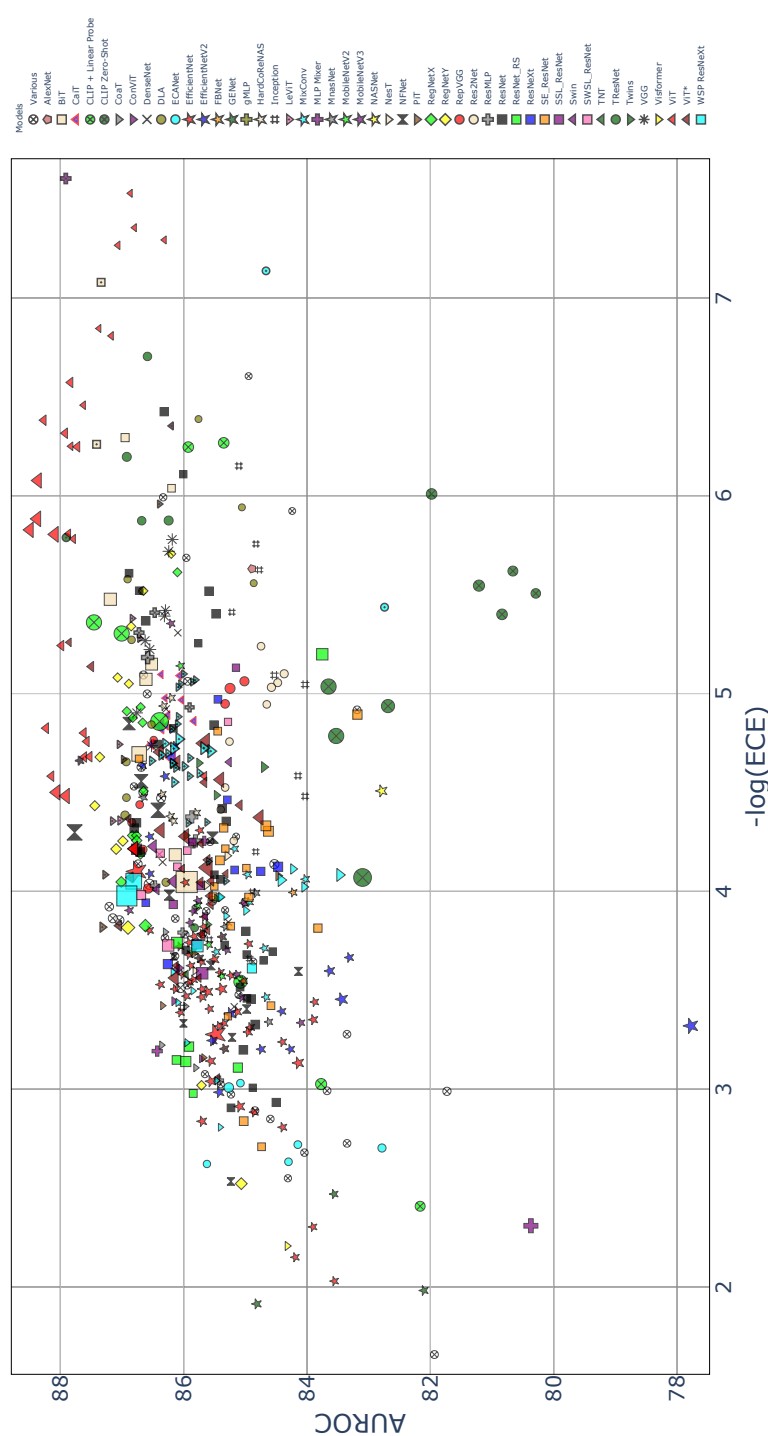

Figure 8: A comparison of 523 models by their AUROC (×100, higher is better) and log(ECE) (lower is better) on ImageNet. Each marker's size is determined by the model's number of parameters. Each dotted marker represents a distilled version of the original. An interactive version of this figure is provided as supplementary material.

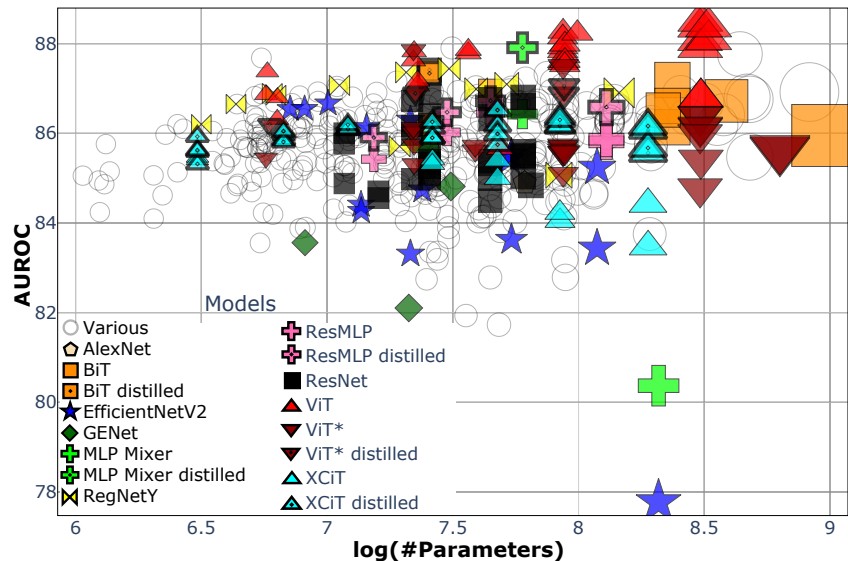

Figure 9: A comparison of 523 models by their AUROC (×100, higher is better) and log(number of model's parameters) on ImageNet. Each dotted marker represents a distilled version of the original.

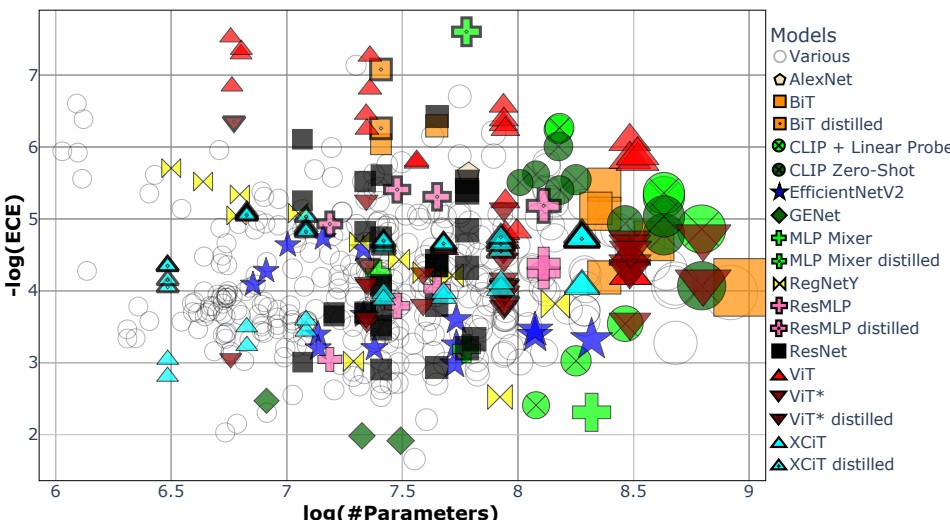

Figure 10: A comparison of 523 models by their -log(ECE) (higher is better) and log(number of model's parameters) on ImageNet. Each dotted marker represents a distilled version of the original.

To demonstrate this in a simple way, let us consider two models with a complete lack of capacity to rank correct and incorrect predictions correctly, always outputting the same confidence score. Model A has an accuracy of 20% (thus an error rate of 80%), and Model B has an accuracy of 80% (and an error rate of 20%). A good ranking metric should evaluate them equally (the same way E-AURC gives the same score for two models that rank perfectly regardless of their accuracy). In Figure 11 we plot their RC curves with dashed lines, which are both straight lines due to their lack of ranking ability. We can calculate both of these models' AURCs, $AURC(modelA) = 0.8, AURC(modelB) = 0.2$.

The next thing to calculate is the best AURC values those models could have achieved given the same accuracy if they had a perfect partial order. We plot these hypothetical models' RC curves in Figure 11 as solid lines. Their selective risk remains 0 for every coverage below their total accuracy, since these hypothetical models assigned the highest confidence to all of their correct instances first. As the coverage increases and they have no more correct instances to select, they begin to give instances that are incorrect, and thus their selective risk deteriorates for higher coverages.

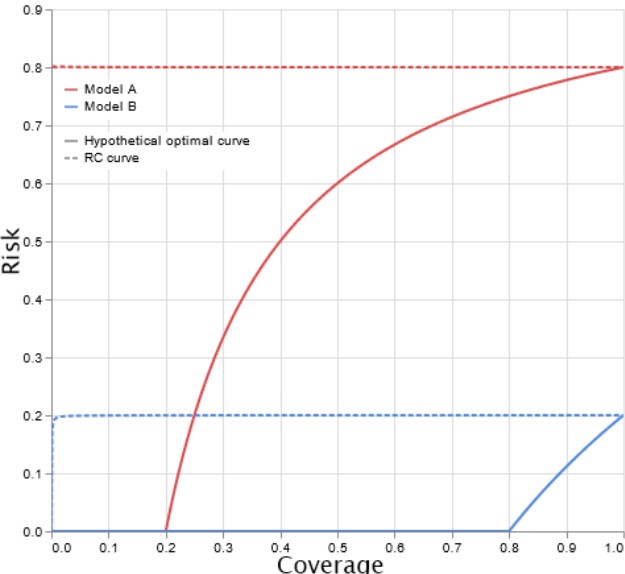

Figure 11: The RC curves for Models A and B are plotted with dashed lines. The RC curves for the hypothetically optimal versions of Models A and B are plotted with solid lines.

Calculating both of these hypothetical models' AURCs gives us the following: $AURC(modelA^*) = 0.482, AURC(modelB^*) = 0.022$. Subtracting our results we get: E-AURC$(modelA) = 0.8 - 0.482 = 0.318$, E-AURC$(modelB) = 0.2 - 0.022 = 0.178$ Hence, E-AURC prefers Model B over Model A, even though both do not discriminate at all between incorrect and correct instances.

## D    MORE ON THE DEFINITION OF RANKING

Let us consider a finite set $S_m = \{(x_i, y_i)\}_{i=1}^{m} \sim P_{X,Y}$. We assume that there are no two identical values given by $\kappa$ on $S_m$. Such an assumption is reasonable when choosing a continuous confidence signal.

We further denote $c$ as the number of concordant pairs (i.e., pairs in $S_m$ that satisfy the condition $[\kappa(x_i, \hat{y}|f) < \kappa(x_j, \hat{y}|f) \cap \ell(f(x_i), y_i) > \ell(f(x_j), y_j)]$) and $d$ as the number of discordant pairs (i.e., pairs in $S_m$ that satisfy the condition $[\kappa(x_i, \hat{y}|f) > \kappa(x_j, \hat{y}|f) \cap \ell(f(x_i), y_i) > \ell(f(x_j), y_j)]$.

We assume, for now, that there are no two identical values given by $\ell$ on $S_m$. Accordingly, we can further develop Equation (1) from Section 2.1 using the definition of conditional probability,

$$\mathbf{Pr}[\kappa(x_i, \hat{y}|f) < \kappa(x_j, \hat{y}|f)|\ell(f(x_i), y_i) > \ell(f(x_j), y_j)] = \\ \frac{\mathbf{Pr}[\kappa(x_i, \hat{y}|f) < \kappa(x_j, \hat{y}|f) \cap \ell(f(x_i), y_i) > \ell(f(x_j), y_j)]}{\mathbf{Pr}[\ell(f(x_i), y_i) > \ell(f(x_j), y_j)]},$$

which can be approximated empirically, using the most likelihood estimator, as

$$\frac{c}{\binom{m}{2}}. \tag{2}$$

We note that the last equation is identical to Kendall's $\tau$ up to a linear transformation, which equals

$$\frac{c - d}{\binom{m}{2}} = \frac{c - d + c - c}{\binom{m}{2}}$$

$$= \frac{2c - (c + d)}{\binom{m}{2}} = \frac{2c}{\binom{m}{2}} - \frac{c + d}{\binom{m}{2}} =$$

$$2 \cdot \frac{c}{\binom{m}{2}} - 1 = 2 \cdot [\text{Equation } 2] - 1.$$

Otherwise, if the loss assigns two identical values to a pair of points in $S_m$, but $\kappa$ does not, then we get:

$$\frac{c}{c + d}. \tag{3}$$

which is identical to Goodman & Kruskal's $\gamma$-correlation up to a linear transformation

$$\frac{c - d}{c + d} = \frac{c - d + c - c}{c + d} = \frac{2c - (c + d)}{c + d} =$$

$$\frac{2c}{c + d} - \frac{c + d}{c + d} = 2 \cdot [\text{Equation } 3] - 1.$$

### D.1 Inequalities of the definition

One might wonder why Equation (1) should have strict inequalities rather than non-strict ones to define ranking. As we discuss below, this would damage the definition:

(1) If the losses had a non-strict inequality:

$$\mathbf{Pr}[\kappa(x_1, \hat{y}|f) < \kappa(x_2, \hat{y}|f)|\ell(f(x_1), y_1) \geq \ell(f(x_2), y_2)]$$

Consequently, in the case of classification, for example, this probability would increase for any pairs consisting of correct instances with different confidences. This would yield no benefit in ranking between incorrect and correct instances and motivates giving different confidence values for instances with the same loss—a fact that would not truly add any value.

(2) If the $\kappa$ values had a non-strict inequality:

$$\mathbf{Pr}[\kappa(x_1, \hat{y}|f) \leq \kappa(x_2, \hat{y}|f)|\ell(f(x_1), y_1) > \ell(f(x_2), y_2)].$$

This probability would increase for any pair $(x_1, x_2)$ such that $\kappa(x_1, \hat{y}|f) = \kappa(x_2, \hat{y}|f)$ and $\ell(f(x_1)) > \ell(f(x_2))$, although $\kappa$ should have ranked $x_1$ with a lower value. Furthermore, if a $\kappa$ function were to assign the same confidence score to all $x \in \mathcal{X}$, then when there are no two identical values of losses, the definition's probability would be 1; otherwise, the more different values for losses there are, the larger the probability would grow. In classification with a 0/1 loss, for example, assigning the same confidence score to all instances would result in the probability being $Accuracy(f) \cdot (1 - Accuracy(f))$, which is largest when $Accuracy(f) = 0.5$.

## E RANKING CAPACITY COMPARISON BETWEEN HUMANS AND NEURAL NETWORKS

In the field of metacognition, interestingly, the predictive value of confidence is evaluated by two different aspects: by its ability to *discriminate* between correct and incorrect predictions (also known as *resolution* in metacognition or ranking in our context) and by its ability to give well-calibrated confidence estimations, not being over- or under-confident (Fiedler et al., 2019). These two aspects correspond perfectly with much of the research done in the deep learning field, with the nearly matching metric to AUROC of $\gamma$-correlation (see Section 2).

This allows us to compare how well humans rank predictions in various tasks versus how well models rank their own in others. Human AUROC measurements in various tasks (translated from

$\gamma$-correlation) tend to range from 0.6 to 0.75 (Undorf & Bröder, 2019; Basile et al., 2018; Ackerman et al., 2016), but could vary, usually towards much lower values (Griffin et al., 2019). In our comprehensive evaluation on ImageNet, AUROC ranged from 0.77 to 0.88 (with the median value being 0.85), and in CIFAR-10 these measurements jump to the range of 0.92 to 0.94.

While such comparisons between neural networks and humans are somewhat unfair due to the great sensitivity required for the task, research that directly compares humans and machine learning algorithms performing the same task exist. For example, in Ackerman et al. (2019), algorithms far surpass even the group of highest performing individuals in terms of ranking.

## F    CRITICISMS OF AUROC AS A RANKING METRIC

In this section, addressing the criticism of AUROC as a ranking metric, we show why AUROC does not simply reward models for having lower accuracy, . The paper by Ding et al. (2019) presented a semi-artificial experiment to demonstrate that AUROC might get larger the worse the model's accuracy becomes. They consider a model $f$ and its $\kappa$ function evaluated on a classification test set $\mathcal{X}$, giving each a prediction $\hat{y}_f(x)$ and a confidence score $\kappa(x, \hat{y}_f(x)|f)$, which in this case is the model's softmax response. Let $\mathcal{X}^c = \{x^c \in \mathcal{X} | \hat{y}_f(x^c) = y(x)\}$ be the set of all instances correctly predicted by the model $f$, and define $x^c_{(i)} \in \mathcal{X}^c$ to be the correct instance that received the i-lowest confidence score from $\kappa$. Their example continues and considers an artificial model $f^m$ to be an exact clone of $f$ with the following modification: for every $i \leq m$, the model $f^m$ now predicts a different, incorrect label for $x^c_{(i)}$; however, its given confidence score remains identical: $\kappa(x^c_{(i)}, \hat{y}_f(x^c_{(i)})|f) = \kappa(x^c_{(i)}, \hat{y}_{f^m}(x^c_{(i)})|f^m)$. $f^0$ is exactly identical to $f$, by this definition, not changing any predictions. The paper shows how an artificially created model $f^m$ obtains a higher AUROC score the bigger its $m$. This happens even though "nothing" changed but a hit to the model's accuracy performance (by making mistakes on more instances).

First, to understand why this happens, let us consider $f^1$: AUROC for $\kappa$ increases the more pairs of $[\kappa(x_1) < \kappa(x_2)|\hat{y}_f(x_1) \neq y(x_1), \hat{y}_f(x_2) = y(x_2)]$ there are. The model $f^1$ is now giving an incorrect classification to $x^c_{(1)}$, but this instance's position in the partial order induced by $\kappa$ has remained the same (since $\kappa(x^c_{(1)})$ is unchanged); therefore, $|\mathcal{X}^c| - 1$ correctly ranked pairs were added: $[\kappa(x^c_{(1)}) < \kappa(x^c_{(i)})|\hat{y}_f(x^c_{(1)}) \neq y(x^c_{(1)}), \hat{y}_f(x^c_{(i)}) = y(x^c_{(i)})]$ for every $1 < i \leq |\mathcal{X}^c|$. Nevertheless, this does not guarantee an increase to AUROC by itself: if, previously, all pairs of (correct,incorrect) instances were ranked correctly by $\kappa$, AUROC would already be 1.0 for $f^0$ and would not change for $f^1$. If AUROC for $f^1$ is higher than it was for $f^0$, this means there exists at least one instance $x^w$ that was incorrectly predicted by the original model $f^0$ such that $\kappa(x^c_{(1)}) < \kappa(x^w)$. Every such *originally* wrongly ranked pair (by $f^0$) of $[\kappa(x^c_{(1)}) < \kappa(x^w)|\hat{y}_f(x^w) \neq y(x^w), \hat{y}_f(x^c_{(1)}) = y(x^c_{(1)})]$ has been eliminated by $f^1$ wrongly predicting $x^c_{(1)}$. This, therefore, causes AUROC to increase at the expense of the model's accuracy.

Such an analysis neglects many factors, which is probably why such an effect is only likely to be observed in artificial models (and not among the actual models we have empirically tested):

1. It is unreasonable to assume that the confidence score given by $\kappa$ will remain exactly the same for an instance $x^c_{(i)}$ given it now has a different prediction. In the case of $\kappa$ being softmax, it assumes the model's logits have changed in a very precise and nontrivial manner. Additionally, by our broad definition of $\kappa$, which allows $\kappa$ to even be produced from an entirely different model than $f$, $\kappa$ receives the prediction and model as a given input (and cannot change or affect either), and it is unlikely to assume changing its inputs will not change its output.

2. Suppose we find the setting reasonable and assume we can actually create a model $f^m$ as described. Let us observe a model $f^p$ such that $p = \min_m(\text{AUROC of } f^m=1)$, meaning that $f^p$ ranks its predictions perfectly, unlike the original $f^0$. Is it really true that $f^p$ has no better uncertainty estimation than $f^0$? Model $f^p$ behaves very much like the investment in "Model B" from our example in Section 1, possessing perfect knowledge of when it is wrong and when it is correct, allowing its users risk-free classification. So, given a model $f$, we can use the above process to produce an improved model $f^p$, and then we can even calibrate its

$\kappa$ to output 0% for all instances below its threshold and 100% for all those above to produce a perfect model, which might have a small coverage but is correct every time, knows it and notifies its user when it truly knows the prediction. The increase in AUROC reflects such an improvement.

Not only do we disagree with such an analysis and its conclusions, but we also have vast empirical evidence to show that AUROC does not prefer lower accuracy models unless there is a good reason for it to do so, as we demonstrate in Figure 3 (comparing EfficientNet-V2-XL to ViT-B/32-SAM). In fact, out of the 523 models we tested, the model with the highest AUROC also has the $4^{th}$ highest accuracy of all models, and the overall Spearman correlation between AUROC and accuracy of all the models we tested is 0.03. Furthermore, Figure 3 also exemplifies why AURC, which was suggested by the just mentioned paper as the alternative to AUROC, is a bad choice as a single number metric, and might lead us to deploy a model that has a worse selective risk for most coverages only due to its higher overall accuracy.

## G    KNOWLEDGE DISTILLATION EFFECTS ON UNCERTAINTY ESTIMATION

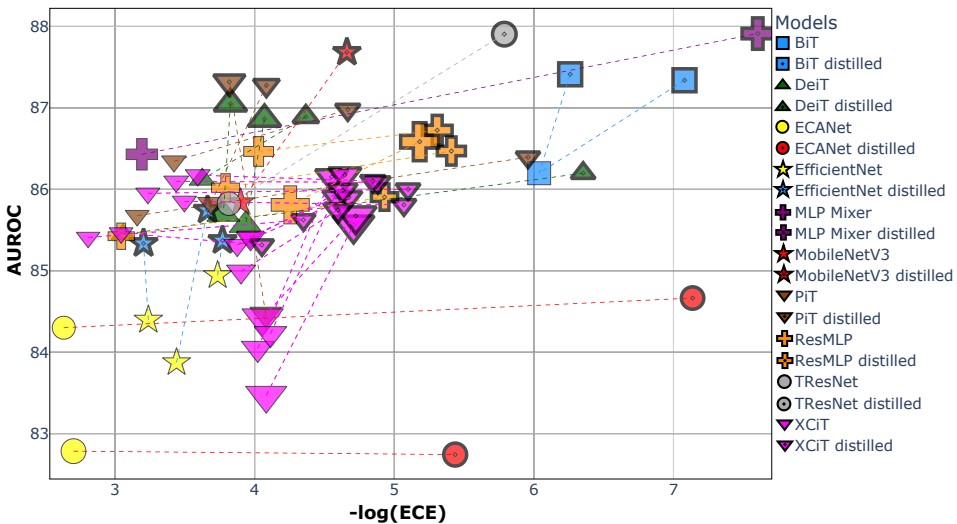

Figure 12: Comparing vanilla models to those incorporating KD into their training (represented by markers with thick borders and a dot). In a pruning scenario that includes distillation, yellow markers indicate that the original model was also the teacher. The performance of each model is measured in AUROC (higher is better) and -log(ECE) (higher is better).

Figure 12 compares vanilla models to those incorporating KD into their training (represented by markers with thick borders and a dot). In a pruning scenario that includes distillation, yellow markers indicate that the original model was also the teacher (Aflalo et al., 2020). While distillation using a different model tends to improve uncertainty estimation in both aspects, distillation by the model itself seems to improve only one—suggesting it is generally more beneficial to use a different model as a teacher. The fact that KD improves the model over its original form, however, is surprising, and implies that the distillation process itself helps uncertainty estimation. Note that although this specific method involves pruning, evaluations of models pruned without incorporating distillation (Frankle & Carbin, 2018) revealed no improvement.

It seems, moreover, that the teacher does not have to be good in uncertainty estimation itself; Figure 5 in Section 3 shows this by comparing the teacher architecture and the students in each case.

While the training method by Ridnik et al. (2021) included pretraining on ImageNet-21k and demonstrated impressive improvements, comparison of models that were pretrained on ImageNet21k (Tan & Le, 2021; Touvron et al., 2021a; 2022) with identical models that were not pretrained showed only a slight improvement in ECE, and, in fact, exhibit a degradation of AUROC (see

Figures 4a and 4b in Section 3). This suggests that pretraining alone does not improve uncertainty estimation.

## H MORE INFORMATION ABOUT TEMPERATURE SCALING

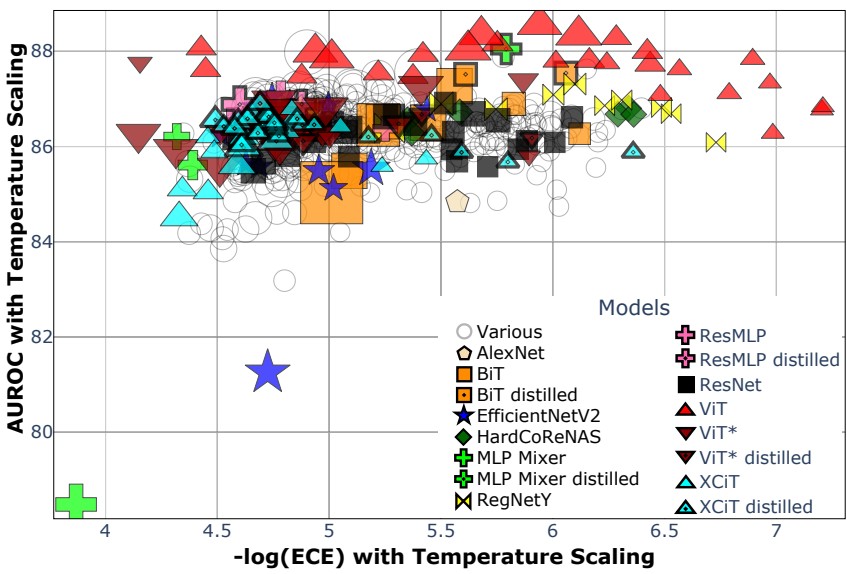

Figure 13: A comparison of 523 models after being calibrated with TS, evaluated by their AUROC (×100, higher is better) and -log(ECE) (higher is better) on ImageNet. Each marker's size is determined by the model's number of parameters. ViT models are still among the best performing architectures for all aspects of uncertainty estimation.

In Figure 13 we see how temperature scaling (TS) affects the overall ranking of models in terms of AUROC and ECE. While the ranking between the different architecture remains similar, the poorly performing models are much improved and minimize the gap between them and the best models. One particularly notable exception is HardCoRe-NAS (Nayman et al., 2021), with its lowest latency versions becoming the top performers in terms of ECE. In addition, models that benefit from

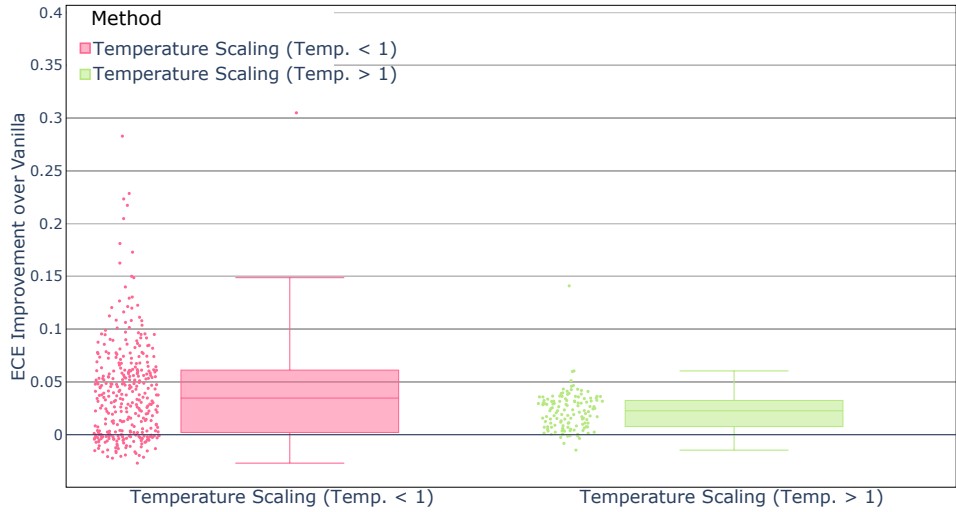

Figure 14: Here the relationship between temperature and the success of TS, unlike the case for AUROC, seems unrelated.

TS in terms of AUROC tend to have been assigned a temperature lower than 1 by the calibration

process (see Figure 6 in Section 3). The same, however, does not hold true for ECE (see Figure 14). This example also emphasizes the fact that models benefiting from TS in terms of AUROC do not necessarily benefit in terms of ECE, and vice versa. Therefore, determining whether to calibrate the deployed model with TS is, unfortunately, a task-specific decision.

We perform TS as was suggested in Guo et al. (2017). For each model we take a random stratified sampling of 5,000 instances from the ImageNet validation set on which to calibrate, and reserve the remainder 45,000 instances for testing. Using the box-constrained L-BFGS (Limited-Memory Broyden-Fletcher-Goldfarb-Shanno) algorithm, we optimize for 5,000 iterations (though fewer iterations usually converge into the same temperature parameter) using a learning rate of 0.01.

## I  ARCHITECTURE CHOICE FOR PRACTICAL DEPLOYMENT BASED ON SELECTIVE PERFORMANCE

As discussed in Section 2, when we know the coverage or risk we require for deployment, the most direct metric to check is which model obtains the best risk for the coverage required (selective risk), or which model gets the largest coverage for the accuracy constraint (SAC). While each deployment scenario specifies its own constraints, for demonstration purposes we consider a scenario in which misclassifications are by far more costly than abstaining from giving correct predictions. An example of this could be classifying a huge unlabeled dataset (or cleaning bad labels from a labeled dataset). While it is desirable to assign labels to a larger portion of the dataset (or to correct more of the wrong labels), it is crucial that these labels are as accurate as possible (or that correctly labeled instances are not replaced with a bad label).

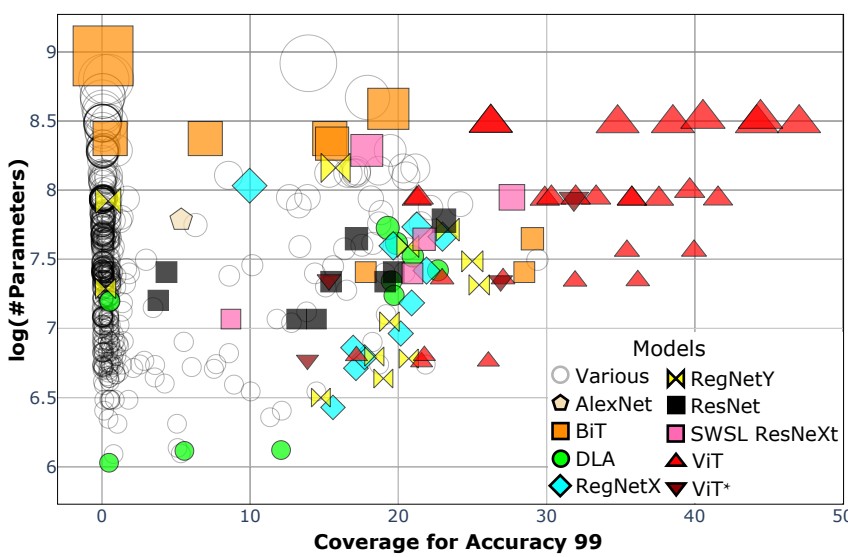

Figure 15: A comparison of 523 models by their log(number of model's parameters) and the coverage they are able to provide for a SAC of 99% (higher is better) on ImageNet.

To explore such a scenario, we evaluate all models on ImageNet to see which ones give us the largest coverage for a required accuracy of 99%. In Figure 7, Section 3 (paper's main body) we observe that of all the models studied, only ViT models are able to provide coverage beyond 30% for such an extreme constraint. Moreover, we note that the coverage they provide is significantly larger than that given by models with comparable accuracy or size, and that ViT models that provide similar coverage to their counterparts do so with less overall accuracy.

In Figure 15 we see that not only do ViT models provide more coverage than any other model, but that they are also able to do so in any size category. To compare models fairly by their size, we present Figure 15, which sets the Y axis to be the logarithm of the number of parameters, so that models sharing the same y value can be compared solely based on their x value—which is the coverage they

Table 1: A comparison of different training regimes of ViTs. *The paper introducing ViTs (Dosovitskiy et al., 2021) had also trained ViT models with the JFT-300M dataset; however, their weights are unavailable to the general public. All evaluations of ViTs from that paper were conducted on ViTs pretrained on ImageNet-21k, which are publicly available. **Pretrained DeiT3 models were first pretrained with a learning rate of $3 \cdot 10^{-3}$ and then fine-tuned with a learning rate of $3 \cdot 10^{-4}$

| Regime | ViT (original) | Steiner et al. | Chen et al. | DeiT | DeiT3 | DeiT3 +Pretraining | Torchvision |
|---|---|---|---|---|---|---|---|
| Reference | Dosovitskiy et al. (2021) | Steiner et al. (2022) | Chen et al. (2022) | Touvron et al. (2021b) | Touvron et al. (2022) | | Paszke et al. (2019) |
| Pretraining dataset | ImageNet-21k* | ImageNet-21k | - | - | - | ImageNet-21k | - |
| Batch Size | 4096 | 4096 | 4096 | 1024 | 2048 | 2048 | 512 |
| Optimizer | AdamW | AdamW | SAM | LAMB | LAMB | LAMB | AdamW |
| LR | $3 \cdot 10^{-3}$ | $3 \cdot 10^{-3}$ | $3 \cdot 10^{-3}$ | $1 \cdot 10^{-3}$ | $3 \cdot 10^{-3}$ | $3 \cdot 10^{-3}$** | $3 \cdot 10^{-3}$ |
| LR decay | cosine | cosine | cosine | cosine | cosine | cosine | cosine |
| Weight decay | 0.1 | 0.3 | 0.1 | 0.05 | 0.02 | 0.02 | 0.3 |
| Warmup epochs | 3.4 | 3.4 | 3.4 | 5 | 5 | 5 | 30 |
| Label smoothing $\epsilon$ | 0.1 | 0.1 | 0.1 | 0.1 | ✗ | 0.1 | 0.11 |
| Dropout | ✓ | ✓ | ✓ | ✗ | ✗ | ✗ | ✗ |
| Stoch. Depth | ✗ | ✓ | ✗ | ✓ | ✓ | ✓ | ✗ |
| Repeated Aug | ✗ | ✗ | ✗ | ✓ | ✓ | ✗ | ✓ |
| Gradient Clip. | 1.0 | 1.0 | 1.0 | ✗ | 1.0 | 1.0 | 1.0 |
| H. flip | ✓ | ✓ | ✓ | ✓ | ✓ | ✓ | ✓ |
| Random Resized Crop | ✓ | ✓ | ✓ | ✓ | ✓ | ✗ | ✓ |
| Rand Augment | ✗ | Adapt. | ✗ | 9/0.5 | ✗ | ✗ | Adapt. |
| 3 Augment | ✗ | ✗ | ✗ | ✗ | ✓ | ✓ | ✗ |
| LayerScale | ✗ | ✗ | ✗ | ✗ | ✓ | ✓ | ✗ |
| Mixup alpha | ✗ | Adapt. | ✗ | 0.8 | 0.8 | ✗ | 0.2 |
| Cutmix alpha | ✗ | ✗ | ✗ | 1.0 | 1.0 | 1.0 | 1.0 |
| Erasing prob. | ✗ | ✗ | ✗ | 0.25 | ✗ | ✗ | ✗ |
| ColorJitter | ✗ | ✗ | ✗ | ✗ | 0.3 | 0.3 | ✗ |
| Test crop ratio | 0.875 | 0.875 | 0.875 | 0.875 | 1.0 | 1.0 | 0.875 |
| Loss | CE | CE | CE | CE | BCE | CE | CE |
| Superb performance | ✓ | ✓ | ✓ | ✗ | ✗ | ✗ | ✗ |

provide for a SAC of 99%. We see that ViT models provide a larger coverage even when compared with models of a similar size.

## J   COMPARISON OF VIT TRAINING REGIMES AND THEIR EFFECTS ON UNCERTAINTY ESTIMATION PERFORMANCE

In Table 1 we compare the different hyperparameters and augmentations used for training the ViT models evaluated in this paper, with the aim of revealing why some training regimes consistently result in superb ViTs, while others do not. An analysis of the various differences between these regimes, however, eliminates the obvious suspects.

1) Pretraining, on its own, does not seem to offer an explanation: First, we analyze eight pairs of models (provided by Touvron et al. 2022) such that both models have identical architecture and training regimes, with the exception that one was pretrained on ImageNet-21k, and the other was not. Pretraining results in only a slight improvement of 0.16 in AUROC and 0.01 in ECE. Moreover, as mentioned in detail in Section 3, ViT models trained on JFT-4B (Tran et al., 2022) were outperformed by the successful ViT models evaluated in this paper, most of which were pretrained on ImageNet-21k (and even by one ViT SAM model that was not pretrained at all). Second, we note that ViTs trained with the SAM optimizer (Chen et al., 2022), and not pretrained at all, reach superb ranking (AUROC) as well. These facts lead us to conclude that pretraining, at least by itself, is not the main contributor to training successful ViTs.

2) The selection of optimizers and other hyperparameters (such as learning rate, label smoothing etc.) does not seem to have a significant impact. For example, while AdamW (Loshchilov & Hutter, 2019) was used by two of the successful regimes, it was also used by Paszke et al. (2019), and on the other hand was replaced by SAM (Foret et al., 2021) in another successful training regime.

3) Advanced augmentations are unlikely to explain the gaps in uncertainty estimation performance, as regimes producing superior ViT models (Dosovitskiy et al., 2021; Chen et al., 2022) did not use advanced augmentations (in comparison to Touvron et al. (2021b) and Touvron et al. (2022), for example).

For these reasons, for the moment, the explanation for the gap remains elusive. The only remaining "suspect" is the batch size used, with all successful regimes using a batch size of 4096, while others use a smaller batch size of 2048 or lower. One could argue, however, that a two-fold increase in batch size is not sufficient to explain the huge gaps in performance measured.

Table 2: The relationship between uncertainty estimation performance and the model's attributes and resources (accuracy, number of parameters and input size), measured by Spearman correlation. Positive correlations indicate good utilization of resources for uncertainty estimation.

| Architecture | AUROC & Accuracy | -ECE & Accuracy | AUROC & #Parameters | -ECE & #Parameters | AUROC & Input Size | -ECE & Input Size | # Models Evaluated |
|---|---|---|---|---|---|---|---|
| EfficientNet | -0.16 | -0.29 | -0.22 | -0.29 | -0.26 | -0.38 | 50 |
| ResNet | -0.28 | -0.22 | 0.16 | 0.03 | -0.40 | -0.44 | 33 |
| ViT | 0.84 | -0.17 | 0.50 | -0.67 | 0.04 | -0.13 | 31 |
| XCiT distilled | 0.60 | 0.09 | 0.35 | 0.02 | 0.51 | 0.12 | 28 |
| XCiT | -0.68 | 0.89 | -0.79 | 0.94 | - | - | 28 |
| ViT* | 0.23 | 0.38 | -0.04 | 0.41 | 0.14 | -0.12 | 26 |
| SE_ResNet | -0.46 | -0.02 | -0.53 | 0.20 | -0.02 | -0.35 | 18 |
| EfficientNetV2 | -0.70 | -0.45 | -0.63 | -0.47 | -0.59 | -0.40 | 15 |
| NFNet | 0.56 | 0.78 | 0.63 | 0.81 | 0.48 | 0.60 | 13 |
| Inception | -0.29 | 0.09 | -0.43 | 0.30 | -0.08 | 0.23 | 13 |
| RegNetY | -0.03 | -0.98 | 0.27 | -0.86 | - | - | 12 |
| RegNetX | 0.20 | -0.96 | 0.20 | -0.96 | - | - | 12 |
| CaiT distilled | 0.44 | -0.87 | 0.35 | -0.87 | 0.58 | -0.50 | 10 |
| DLA | 0.64 | -0.90 | 0.77 | -0.90 | - | - | 10 |
| MobileNetV3 | 0.37 | 0.59 | 0.42 | 0.60 | - | - | 10 |
| Res2Net | -0.70 | 0.27 | -0.68 | 0.60 | - | - | 9 |
| CLIP Zero-Shot | 1.0 | -0.63 | 0.9 | -0.8 | 0.55 | -0.58 | 9 |
| CLIP + Linear Probe | 0.88 | 0.26 | 0.71 | 0.1 | 0.19 | -0.27 | 8 |
| VGG | 0.81 | -0.98 | 0.71 | -0.90 | - | - | 8 |
| RepVGG | -0.71 | 0.50 | -0.57 | 0.21 | - | - | 8 |
| BiT | -0.33 | -0.81 | -0.20 | -0.85 | -0.46 | -0.25 | 8 |
| ResNeXt | -0.96 | 0.39 | -0.22 | -0.30 | - | - | 7 |
| ResNet RS | 0.00 | 0.79 | -0.18 | 0.82 | -0.30 | 0.82 | 7 |
| MixConv | -0.11 | 0.89 | -0.24 | 0.86 | - | - | 7 |
| DenseNet | 0.43 | -0.14 | 0.72 | 0.12 | - | - | 6 |
| HardCoReNAS | -0.60 | 0.26 | -0.49 | 0.37 | - | - | 6 |
| Swin | 0.71 | 0.14 | 0.77 | 0.26 | 0.41 | 0.00 | 6 |
| ECANet | -0.20 | 0.60 | -0.43 | 0.37 | 0.83 | 0.37 | 6 |
| Twins | -0.26 | 0.94 | -0.14 | 0.89 | - | - | 6 |
| SWSL ResNet | 0.94 | -0.89 | 0.77 | -0.83 | - | - | 6 |
| GENet | 0.50 | -1.00 | 0.50 | -1.00 | 0.87 | -0.87 | 6 |
| SSL ResNet | 0.14 | -1.00 | 0.26 | -0.94 | - | - | 6 |
| TResNet | 0.10 | -0.30 | 0.53 | 0.53 | -0.58 | -0.87 | 5 |
| CoaT | -0.10 | 0.90 | -0.10 | 0.50 | - | - | 5 |
| LeViT distilled | 0.60 | -0.90 | 0.60 | -0.90 | - | - | 5 |
| ResMLP | 0.20 | 1.00 | 0.15 | 0.97 | - | - | 5 |
| MobileNetV2 | -0.30 | 0.00 | -0.21 | 0.10 | - | - | 5 |
| ViT* Distilled | 0.8 | -1.0 | 0.71 | -0.77 | 0.22 | -0.77 | 4 |
| PiT distilled | 1.00 | -1.00 | 1.00 | -1.00 | - | - | 4 |
| PiT | -0.40 | 1.00 | -0.40 | 1.00 | - | - | 4 |
| WSP ResNeXt | 1.00 | 0.80 | 1.00 | 0.80 | - | - | 4 |
| ResMLP distilled | 0.80 | 0.20 | 0.80 | 0.20 | - | - | 4 |
| MnasNet | 0.40 | 0.20 | 0.63 | 0.95 | - | - | 4 |

## K  EVALUATIONS OF THE ZERO-SHOT LANGUAGE–VISION CLIP MODEL

In this section we describe how we use CLIP model and extract confidence signals from it during inference. To evaluate CLIP on ImageNet, we first prepare it following the code provided by its authors (https://github.com/openai/CLIP): The labels of ImageNet-1k are encoded into normalized embedding vectors. At inference time, the incoming image is encoded into another normalized embedding vector. A cosine similarity is then calculated between each label embedding vector and the image embedding vector, and lastly, softmax is applied. The highest score is then taken as the confidence score for that prediction. We also evaluate the same models when adding a trained "linear-probe" to them (as described in Radford et al. (2021), which is essentially a logistic regression head), that results in a large boost in their accuracy.

## L  EFFECTS OF THE MODEL'S ACCURACY, NUMBER OF PARAMETERS AND INPUT SIZE ON UNCERTAINTY ESTIMATION PERFORMANCE

Table 2 shows the relationship between uncertainty estimation performance and model attributes and resources (accuracy, number of parameters and input size), measured by Spearman correlation. We measure uncertainty estimation performance by AUROC (higher is better) and -ECE (higher is better). Positive correlations indicate good utilization of resources for uncertainty estimation (for example, a positive correlation between -ECE and the number of parameters indicates that as the number of parameters increases, the calibration improves). An interesting observation is that distillation can drastically change the correlation between a resource and the uncertainty estimation performance metrics. For example, undistilled XCiTs have a Spearman correlation of -0.79 between their number of parameters and AUROC, indicating that more parameters are correlated with lower ranking performance, while distilled XCiTs have a correlation of 0.35 between the two.

Table 3: Comparing using MC dropout to softmax-response (vanilla).

| Architecture | Method | Accuracy | AUROC |
|---|---|---|---|
| MobileNetV3 Large | Vanilla | **74.04** | **86.88** |
| | MC dropout | 74 | 86.14 |
| MobileNetV3 Small | Vanilla | **67.67** | **86.2** |
| | MC dropout | 67.55 | 84.54 |
| MobileNetV2 | Vanilla | **71.88** | **86.05** |
| | MC dropout | 71.81 | 84.68 |
| VGG11 | Vanilla | **70.37** | **86.31** |
| | MC dropout | 70.21 | 84.3 |
| VGG11 (no BatchNorm) | Vanilla | **69.02** | **86.19** |
| | MC dropout | 68.95 | 83.94 |
| VGG13 | Vanilla | **71.59** | **86.3** |
| | MC dropout | 71.43 | 84.37 |
| VGG13 (no BatchNorm) | Vanilla | **69.93** | **86.24** |
| | MC dropout | 69.71 | 84.3 |
| VGG16 | Vanilla | **73.36** | **86.76** |
| | MC dropout | 73.33 | 85.02 |
| VGG16 (no BatchNorm) | Vanilla | **71.59** | **86.63** |
| | MC dropout | 71.47 | 84.97 |
| VGG19 | Vanilla | **74.22** | **86.52** |
| | MC dropout | 74.17 | 85.06 |
| VGG19 (no BatchNorm) | Vanilla | **72.38** | **86.55** |
| | MC dropout | 72.37 | 84.99 |

## M    EVALUATIONS OF MONTE CARLO DROPOUT RANKING PERFORMANCE

MC Dropout (Gal & Ghahramani, 2016) is computed using several dropout-enabled forward passes to produce uncertainty estimates. In classification, the mean softmax score of these passes, is calculated, and then a predictive entropy score is used as the final uncertainty estimate. In our evaluations, we use 30 dropout-enabled forward passes. We do not measure MC Dropout's effect on ECE since entropy scores do not reside in $[0, 1]$.

We test this technique using MobileNetV3 (Howard et al., 2019), MobileNetv2 (Sandler et al., 2018) and VGG (Simonyan & Zisserman, 2015), all trained on ImageNet and taken from the PyTorch repository (Paszke et al., 2019).

The results comparing these models with and without using MC dropout are provided in Table 3.

The table shows that using MC dropout causes a consistent drop in both AUROC and selective performance compared with using the same models with softmax as the $\kappa$. These results are also visualized in comparison to other methods in Figure 4a in Section 3. MC dropout underperformance in an ID setting was also previously observed in Geifman & El-Yaniv (2017).

