# OpenReview forum: "What Can we Learn From The Selective Prediction And Uncertainty Estimation Performance Of 523 Imagenet Classifiers?"
_ICLR.cc/2023/Conference — ICLR 2023 poster_

### Official Review · Reviewer_CSk1 · 2022-10-24

**Confidence:** 4
**Clarity, Quality, Novelty And Reproducibility:** See above
**Correctness:** 3
**Technical Novelty And Significance:** 2
**Empirical Novelty And Significance:** 4
**Recommendation:** 6

**Strength And Weaknesses:**

From my understanding, this work relies on trained models released by various previous works, which precludes the authors from controlling optimization parameters such as learning rate, regularization parameters, and training time; these hyperparameters may have significant impact on calibration due to their effects on overfitting. Furthermore, the analysis in this work is limited to imagenet (and 1 plot with cifar 10?), which may limit the transferability of the observations made in this paper.

The way in which results are presented in the paper is also problematic at times. For example, the box plots in figure 4 and figure 5 did not include the number of pairs in each column, thereby limiting the reader’s ability to interpret the statistical significance of the results illustrated. In another example, while the authors repeatedly stated that they evaluated 523 models, not a figure, table, or graph in the supplementary materials explicitly lists what these 523 models consist of. For the sake of reproducibility, a machine-readable but human friendly list of models should accompany the supplementary materials (so that readers can see which models are used without prowling the html source). Since the authors have already grouped these models into tens of categories for plotting (different marker types in figure 1), tables summarizing the various models within each category would be very informative in the appendix. Furthermore, a distilled version of table 2 with the number of models within each category would be beneficial to understanding in the main text. The paper is also not fully self-contained; a review of related works that measure and benchmark uncertainty estimation and calibration is missing. This makes it difficult to discern which results in this paper are in accordance with current knowledge, and which results are surprising. Also missing are descriptions of the different methods applied to the models (e.g. adversarial training, KD, temperature scaling etc.). A concise and mathematical description of each method should at least be included in the appendix. The omission of this information diminishes the persuasiveness of the paper’s message.

Nonetheless, this work presents a trove of quantitative results from which interesting conclusions can be drawn. The six conclusions drawn from these results are salient and actionable. As such, I would recommend for acceptance, as I believe that the publication of this work will benefit the collective knowledge of our field.

One question I have is whether the observations from this work can transfer to any other dataset, using a more selective subset of models that had shown significant variation in performance on imagenet.

Other minor notes:
There’s a kappa on page two that is not rendered correctly as the symbol.
Figure placement and captioning can be significantly tightened (e.g. figure 4 and 5 has basically the same caption).

**Summary Of The Paper:**

This work examines the uncertainty calibration of various image classifiers. A fairly comprehensive list of classification models, calibration techniques, and metrics are examined experimentally. From experimental results, this work draws several observations and recommendations in both model type and training techniques.


**Summary Of The Review:**

See above

---

> ### Author Response · Authors · 2022-11-07
> **Authors' response**
>
> Thank you for your constructive feedback,
>
> We appreciate your suggestions and agree they will improve our paper and will work to implement them in our revision.
>
> We liked your idea to add a list containing the models we have evaluated to the supplementary material. In fact, we already include an initial version of this list in our first revision.
>
> *"One question I have is whether the observations from this work can transfer to any other dataset, using a more selective subset of models that had shown significant variation in performance on imagenet."*
>
> From our experience, ImageNet results tend to transfer.
> Thus, we believe that many of the conclusions do transfer to other classification tasks.
> However, we agree that this is a good research question.

---

### Official Review · Reviewer_GVar · 2022-10-25

**Confidence:** 4
**Correctness:** 4
**Technical Novelty And Significance:** 2
**Empirical Novelty And Significance:** 3
**Recommendation:** 6

**Clarity, Quality, Novelty And Reproducibility:**

Clarity: The paper is clear.
Novelty: The presents very interesting empirical results but lacks algorithmic novelty
Reproducibility: It is not clear if the code will be released

**Strength And Weaknesses:**

Strength:
- The paper is well-written
- The insights presented are very interesting
- The experiments are extensive including several metrics and pretrained models
- The community can benefit from the presented results by investigating why some architectures have low uncertainty and this can open the door designing better methods.

Weaknesses:
- The main point is the lack of algorithmic novelty

**Summary Of The Paper:**

The paper presents an extensive empirical study on the selective prediction and uncertainty estimation. The experiments include a large number of pretrained models and several metrics. Although there is no novel algorithmic novelty, the results presented are very intersting.

**Summary Of The Review:**

Overall, the paper presents an interesting empirical study on selective classification and uncertainty estimation. Although the paper lacks algorithmic novelty, the results presented
It could have been better if the authors analyzed one of the presented insights and tried to find intuition for it.

---

> ### Author Response · Authors · 2022-11-07
> **Reproducibility**
>
> Thanks for your feedback,
>
> We will of course release our code. That along with the fact that all models are publicly available will make our paper very easy to reproduce.

---

### Official Review · Reviewer_a33z · 2022-10-25

**Confidence:** 4
**Correctness:** 3
**Technical Novelty And Significance:** 3
**Empirical Novelty And Significance:** 3
**Recommendation:** 5

**Clarity, Quality, Novelty And Reproducibility:**

- This work is well written. It clearly introduces the involved uncertainty estimation metrics. Moreover, the analysis seems interesting.

- The models are provided by TIMM and Pytorch, so this work is reproducible.

**Strength And Weaknesses:**

**[Strength]**
- Studying uncertainty estimation of deep models is important
- This work considers commonly used uncertainty metrics. Also, diverse models with various architectures, training regimes, and learning schedules are included in the study.

**[Weaknesses]**

- The major question is: this work conduct analysis on ImageNet validation set, which is an in-distribution dataset. When testing models on ***out-of-distribution (OOD)*** datasets (e.g., ImageNet-R, ImageNet-S, and ObjectNet), the observations in this work ***might not hold***. In real-world applications, the dataset distribution typically changes. Thus, I would like to see the study on OOD datasets. Moreover, the models with the same accuracy can have significantly different performances on OOD datasets. Then, in addition to Figures 3 and 4, reporting the accuracy of models on OOD datasets would be more helpful to understand the effect of different methods.

- Section 3 uses a condition that the models have the same classification accuracy. This seems reasonable. How about different architectures with the same accuracy? I would like to check the effect of architecture on uncertainty estimation accuracy.

- Figure 6 (Comparing teacher models to their KD students ) is a little confusing. The student models have different accuracy from their teachers. In this case, it might not reasonable to compare them. Please clarify this.

- Please clarify on which dataset the temperature scaling was learned.


**Summary Of The Paper:**

This work studies the ***relationship*** between ***deep models*** and their corresponding ***selective prediction and uncertainty estimation performance***. Specifically, this work considers several uncertainty estimation metrics, including AUROC, ECE, AURC and coverage for selective accuracy constraint. Using 523 models, this work provides some observations. For example, distillation-based training regimes consistently yield better uncertainty estimations; a subset of ViT models that outperform any other models in terms of uncertainty estimation performance.

**Summary Of The Review:**

**I am around the borderline**.
- Overall, this work is well-written and easy to follow. The study on the relationship between deep models and uncertainty estimation performance is useful.
- While the analysis is interesting, I expect to see the observations on out-of-distribution datasets.
- Some observations need more clarification (e.g., Figure 6).

***------Post-Rebuttal------***

The replies did not answer initial concerns. I have been waiting for the required clarifications. If not offered, I would reject this paper and be willing to fight for my rating.

---

> ### Author Response · Authors · 2022-11-07
> **Evaluating results on OOD**
>
> Thank you, we appreciate your feedback,
>
> *"The major question is: this work conduct analysis on ImageNet validation set, which is an in-distribution dataset. When testing models on out-of-distribution (OOD) datasets (e.g., ImageNet-R, ImageNet-S, and ObjectNet), the observations in this work might not hold. In real-world applications, the dataset distribution typically changes. Thus, I would like to see the study on OOD datasets. Moreover, the models with the same accuracy can have significantly different performances on OOD datasets. Then, in addition to Figures 3 and 4, reporting the accuracy of models on OOD datasets would be more helpful to understand the effect of different methods."*
>
> We agree that the paper would greatly benefit from additional experiments, and in particular on OOD datasets.
> While the paper is presently loaded with many empirical results (and it would be very hard to find space for more), we will make our best effort to squeeze these experiments in the remaining rebuttal time.
> Having said that, we already have results for OOD detection of samples that have unknown OOD labels (such as the ImageNet-O dataset [1]).
> In summary, the conclusions from this study remain qualitatively the same. Specifically, knowledge distillation also significantly improves performance on OOD detection (a different setting than the one considered in this paper), and ViTs remain the top-performing architectures. The most significant difference between the results is that, unlike the ID case, MC dropout does improve performance on OOD detection (a fact we briefly mention in the last paragraph of Section 3, in the point concerning MC dropout).
>
> Thanks a lot for all of your other comments. We will make sure to clarify these points in the revision.
>
> [1] Hendrycks et. al, Natural Adversarial Examples, CVPR 2021

---

> > ### Comment · Reviewer_a33z · 2022-11-20
> > **Need clarification**
> >
> > Dear Authors:
> >
> > Please reply to my questions/weaknesses in my initial reviews. The current reply did not address them.
> >
> > Moreover, the out-of-distribution dataset means the dataset exhibits a distribution shift with the training set. For example, the commonly used datasets are ImageNet-R, ImageNet-S, and ObjectNet. I did mean the dataset for the task of out-of-distribution detection where the class label space is totally different. But thank you for mentioning ImageNet-OOD.
> >
> >
> >
> > Best,
> >
> > Reviewer a33z

---

> > > ### Author Response · Authors · 2022-12-08
> > > **Authors' Response**
> > >
> > > We are terribly sorry for our late response. We were unable to respond due to a personal issue.
> > >
> > > It is very common in the literature to compare uncertainty estimation performance between models with different accuracy levels. For example, [1,2,3,4,5] to name a few. There are plenty more.
> > >
> > > We also find this reasonable and discuss some metrics we find less appropriate for such comparisons (in Section 2 and Appendix C, we demonstrate that AURC and its direct derivatives are less suitable).
> > >
> > > A good example is shown in Figure 3, in which two models have a top-1 accuracy gap of 12% (EfficientNet-V2-XL and ViT-B/32-SAM). Despite this, the less accurate model exhibits better performance when it comes to AUROC, SAC, and selective risk (with the exception of selective risk for high coverage rates).
> > >
> > > This is why we made a comparison between the teacher models and their KD students.
> > > We are aware that getting the point across is essential for clarity. Do you think we should add a summary of our short discussion to the revision to clarify this?
> > >
> > > *"How about different architectures with the same accuracy? I would like to check the effect of architecture on uncertainty estimation accuracy."*
> > >
> > > An example of such a comparison can be seen in Figure 7, using the SAC metric. Different architectures are compared for equal accuracy values (represented by the y-axis). You can also view this figure in an interactive Plotly version ("sac_accuracy.html") for a more comfortable view.
> > >
> > > Furthermore, we have provided a table of all raw results in the supplementary ("results.xlsx"). By controlling the "accuracy" column, you may compare models with similar accuracy values and any other uncertainty metric we have evaluated.
> > >
> > > *"Please clarify on which dataset the temperature scaling was learned."*
> > >
> > > We perform temperature scaling as suggested by [1]. For each model we take a random stratified sampling of 5,000 instances from the ImageNet validation set on which to calibrate, and reserve the remainder 45,000 instances for testing. Using the box-constrained L-BFGS algorithm, we optimize for 5,000 iterations (though fewer iterations usually converge into the same temperature parameter) using a learning rate of 0.01.
> > >
> > > Appendix H ("More information about temperature scaling") already mentions this, and we will make sure to add a reference to it in the main paper in our next revision.
> > >
> > > Thank you again for your feedback.
> > >
> > > [1] Guo, et al. On calibration of modern neural networks, ICML 2017.
> > >
> > > [2] Krishnan & Tickoo, Improving model calibration with accuracy versus uncertainty optimization, NeurIPS 2020.
> > >
> > > [3] Geifman, et al. Bias-reduced uncertainty estimation for deep neural classifiers, ICLR 2019.
> > >
> > > [4] Ziyin, et al. Deep Gamblers: Learning to abstain with portfolio theory, NeurIPS 2019.
> > >
> > > [5] Tran, et al. Plex: Towards Reliability using Pretrained Large Model Extensions, ICML 2022.

---

### Official Review · Reviewer_5WoK · 2022-11-02

**Confidence:** 5
**Correctness:** 2
**Technical Novelty And Significance:** 2
**Empirical Novelty And Significance:** 3
**Recommendation:** 8

**Clarity, Quality, Novelty And Reproducibility:**

About clarity and quality, I believe this is high quality work and it is mostly clear clear, there are some issues in presentation of results (missing information that I mention in weaknesses and minor issues) that hinder the interpretability of results. There is a good chunk of missing information like an exact list of the models architectures that were tried for each experiment.

About novelty, the paper has some novel aspects, in particular findings about uncertainty (minus my comment about out of distribution detection) for vision transformers are novel, the analysis of risk/coverage also I believe is novel, but the largest novel contribution is the analysis of different training regimes, which has a strong conclusion that knowledge distillation improves uncertainty quantification considerably, and temperature scaling also has an important effect, both on ECE and AUROC.
There are some claims that are not novel or well known knowledge from statistics, like that ECE is not a proper measure for model selection and there are ways to trick it.

Reproducibility of this paper is slightly weak, the authors use many pretrained models from a clear source, but some plots are not reproducible due to lack of information about which exact models were used to produce those plots, I document these issues above.


**Strength And Weaknesses:**

Strengths
- The paper is very well written and presented, I have no complaints about writing or presentation except some details on minor issues.
- It makes sense to make a large scale analysis of pretrained models on ImageNet about their (uncalibrated) uncertainty quantification capabilities, specially considering the recent advances in the last years with vision transformers. While I have my reservations about metrics and models with uncertainty, at least this is not a bad idea to start with.
- I like the analysis of risk-coverage, I have not seen this before at this level of depth, with so many models, and I believe this could be a novelty in this paper. The conclusion here is that some vision transformers produce better ranking and selective prediction, as measured by the coverage-accuracy plot.
- I like the conclusion that some vision transformers have better quality of uncertainty, but I think this is also not new since there is previous research that shows vision transformers improve on out of distribution detection, which is also closely related to uncertainty estimation. For this you can view the paper "Exploring the Limits of Out-of-Distribution Detection" by Stanislav Fort at NeurIPS 2021. I do not think this completely invalidates the results but at least it is something to acknowledge.
- One of the major results of this paper is an evaluation of different training regimes, including adversarial training, pretraining on ImageNet21K, semi-supervised learning, distillation, temperature scaling, and MC-dropout, on different uncertainty quantification metrics. Distillation seems to have the largest impact in improving AUROC and ECE over their standard variations. I believe this is a novel finding as I think it is not known at this scale.
- Additionally, temperature scaling also has a good improvement on AUROC and ECE and since it is easy to implement, it is a worthy information for practitioners to try to improve basic uncertainty quantification of their models.
- A good conclusion is that some particular vision transformers architectures (like the original ViT) clearly outperform all other architectures in terms of accuracy vs coverage, which I also think it is a novel finding.
- Another good analysis is correlation between AUROC and ECE with other parameters, they could be positive, negative, or zero, so not many conclusions can be drawn from this. Note that I have some criticism in weaknesses about this, specially since correlation is linear and there could be non-linear effects between these metrics.

Weaknesses
- The major problem with this paper is that all the pretrained models evaluated in this paper have been trained only with empirical risk minimization (except the ones with alternate training methods), meaning they are not really tuned to produce high quality uncertainty. I recognize that the authors used some models that use monte carlo dropout, but this is a very basic method for uncertainty estimation, a much better comparison would have been using bayesian neural networks and similar methods aimed to produce high quality uncertainty (ensembles, flipout, etc).
- The argument that ECE does not always point to the best model is well known, the ECE can be zero in degenerate cases like predicting mean accuracy as confidence, the ECE is not a proper scoring rule, so I think this is argument is not something new and severely weakens the paper. There are other calibration concepts that could have been used instead like group or adversarial calibration, for this see https://arxiv.org/abs/2006.10288

~~- In Figures 4 and 5, there is a large variation in the number of samples for each category of comparison (adv training, semi supervised, ImageNet21K, etc), with temperature scaling having the largest number of samples by far, I think the paper should comment on this as the sample size has an effect on the quality of the comparison and the conclusions that follow it, ideally all methods should use similar samples. Here I also assume that each data point is one different model being used, the caption does not make this clear. And there seems to be no explanation why different number of models were used in each of these experiments, particularly for MC-dropout for example.~~

~~- In Section 3, subpoint 4, the authors mention correlation of AUROC/ECE with several metrics are zero. The issue is the claim that this contradicts previous smaller scale studies, to me it is not clear which studies, and which exact claims. The paper later cites [Guo et al. 2017], but this paper does not evaluate AUROC, as that paper is about calibration and uses ECE for most experiments, so it is not clear which study claims to have found other correlations between AUROC and model parameters. This needs to be strongly clarified.~~

~~- While I understand the idea of using the AUROC to gauge ranking of predicted probabilities, the paper makes references to AUROC for binary classification (this is the most common use), but the results are for multi-class classification, and the paper does not mention which formulation is used for this case, as it is not trivial to extend binary classification AUROC into multi-class AUROC, so I would expect that the authors describe this in some level of detail, currently it is completely missing.~~

Minor Issues
- For Reproducibility, I would expect a list or at least some summary or information on which ones are the 523 models used in the evaluation of this paper, the authors refer to versions of the timm library but from the documentation (at https://rwightman.github.io/pytorch-image-models/models/ ) it is not clear where the figure of 523 models comes from.
- In Figure 8, what is the "various" models plotted with circles? This is not clear from the caption and should be clarified.

**Summary Of The Paper:**

This paper is about quality of confidence predictions made by a bunch of pretrained models on ImageNet 1K, in terms of selective prediction (measured by the accuracy vs confidence plot), ranking (measured by AUROC), and calibration (using ECE).

The authors claim to evaluate 523 pretrained models from PyTorch and timm in terms of the above mentioned metrics, and explore issues like ranking and selective prediction performance (where a classifier can abstrain from making a prediction if its uncertainty is too high). There are some important conclusions and insights that can be drawn from this study that are summarized below and in the strengths.

The contributions are:
- A large scale study of pretrained models on ImageNet 1K in terms of the quality of their predictive uncertainty.
- Conclusion that knowledge distillation improves uncertainty estimation best as compard with other training regimes.
- Conclusion that temperature scaling also improves AUROC, meaning better selective and ranking performance.
- Finding that some vision transformer architectures, particularly ViT, have better uncertainty estimation than other comparable architectures.
- Information about large scale correlations between AUROC, ECE, and model parameters like accuracy, number of parameters. There are good conclusions that not much can be inferred in terms of correlations from these metrics.


**Summary Of The Review:**

I have mixed feelings about this paper, in one side there are some good contributions and conclusions, this is a large scale study which can have a major impact in future research, specially since it uses ImageNet and many models across different architectural patterns, but in the other side there are some known or dubious claims that weaken the paper, but the major issue is the lack of information and clarity about how the experiments were performed, particularly in some figures, and selection of models for each experiment. This paper has potential but needs some strong clarifications (these are documented in weaknesses and minor issues).

~~I believe that right now this paper is borderline, this could definitely change with author feedback. My major issue is clarification about claims, previous work, and definition of AUROC.~~

After the rebuttal, I can recommend this paper for acceptance if the clarifications made by the authors during rebuttal are implemented in the final version.

---

> ### Author Response · Authors · 2022-11-07
> **Authors' response p1**
>
> Your highly detailed and informative feedback is appreciated,
>
> *"The argument that ECE does not always point to the best model is well known, the ECE can be zero in degenerate cases like predicting mean accuracy as confidence, the ECE is not a proper scoring rule, so I think this is argument is not something new and severely weakens the paper. There are other calibration concepts that could have been used instead like group or adversarial calibration"*
>
> We are aware of the criticisms regarding ECE, and in the last paragraph of Section 2 we acknowledge and cite previous work and alternatives to ECE. The primary reason we considered ECE is its overwhelming popularity.
> The Zhao, Ma and Ermon paper you mentioned clearly offers an interesting approach to calibration and that should be mentioned in our paper.
>
> We emphasize that we do not presume to contribute anything novel to the discussion of which metric should be used to measure calibration properly. On the contrary, after presenting the investment example (in the introduction) we state that it is easy to construct examples for which ECE would succeed and others (e.g., AUROC and selective risk) fail.
> Thus, the purpose of the investment example is to stress the importance of considering many different performance metrics to encourage future research to do the same.
>
> The severity of your comment on this issue clearly shows that our exposition failed to deliver our original intent across, and instead gave the wrong impression that we take a stance. We will appreciate it if you could point out the particular phrasing in our paper that raises this issue.
>
> *"Another good analysis is correlation between AUROC and ECE with other parameters, they could be positive, negative, or zero, so not many conclusions can be drawn from this. Note that I have some criticism in weaknesses about this, specially since correlation is linear and there could be non-linear effects between these metrics."*
>
> We have used Spearman rank correlation in our analysis (i.e., a non-linear ranking correlation).
>
> *"In Figures 4 and 5, there is a large variation in the number of samples for each category of comparison (adv training, semi supervised, ImageNet21K, etc), with temperature scaling having the largest number of samples by far, I think the paper should comment on this as the sample size has an effect on the quality of the comparison and the conclusions that follow it, ideally all methods should use similar samples. Here I also assume that each data point is one different model being used, the caption does not make this clear. And there seems to be no explanation why different number of models were used in each of these experiments, particularly for MC-dropout for example."*
>
> The number of models in each sample was affected by both the supply of models, and the amount of compute each method might require.
> For example, the application of temperature scaling is computationally cheap so we were able to apply it to all the models we evaluated.
> In regards to training regimes (which we do not control), we simply took all models that were available with those regimes.
> We agree that this issue should be clarified and we plan to do so.
>
> *"In Section 3, subpoint 4, the authors mention correlation of AUROC/ECE with several metrics are zero. The issue is the claim that this contradicts previous smaller scale studies, to me it is not clear which studies, and which exact claims. The paper later cites [Guo et al. 2017], but this paper does not evaluate AUROC, as that paper is about calibration and uses ECE for most experiments, so it is not clear which study claims to have found other correlations between AUROC and model parameters. This needs to be strongly clarified."*
>
> You are correct. That study considered ECE and our findings only contradict it regarding ECE's correlations. We will strongly clarify this point in the revision.
>
> *"While I understand the idea of using the AUROC to gauge ranking of predicted probabilities, the paper makes references to AUROC for binary classification (this is the most common use), but the results are for multi-class classification, and the paper does not mention which formulation is used for this case, as it is not trivial to extend binary classification AUROC into multi-class AUROC, so I would expect that the authors describe this in some level of detail, currently it is completely missing."*
>
> We certainly agree that this is not trivial. That's the reason we explained it and formulated it in Equation (1) (presented in the first paragraph of Section 2.1), with the loss being a 0/1 loss.
> While the classifier itself is a multi-class classifier (as you noted), the objective of its confidence function is to rank predictions by their correctness, which is a binary classification (the prediction is either correct or incorrect).

---

> > ### Author Response · Authors · 2022-11-07
> > **Authors' response p2**
> >
> > *"I like the conclusion that some vision transformers have better quality of uncertainty, but I think this is also not new since there is previous research that shows vision transformers improve on out of distribution detection, which is also closely related to uncertainty estimation. For this you can view the paper "Exploring the Limits of Out-of-Distribution Detection" by Stanislav Fort at NeurIPS 2021. I do not think this completely invalidates the results but at least it is something to acknowledge."*
> >
> > Thanks for pointing out this paper, it could help strengthen our conclusion. Our revision will include it.
> >
> > *"For Reproducibility, I would expect a list or at least some summary or information on which ones are the 523 models used in the evaluation of this paper, the authors refer to versions of the timm library but from the documentation (at https://rwightman.github.io/pytorch-image-models/models/ ) it is not clear where the figure of 523 models comes from."*
> >
> > We value the complete reproducibility of our paper and will make sure that every detail in it paper will be reproducible.
> >
> > 1) The supplementary material already contains our figures as interactive plotly plots, and provides information about every single data point (model).
> >
> > 2) We now also include a table of the models along with their results in the supplementary material.
> >
> > 3) Upon acceptance we will release our code.
> >
> > *"In Figure 8, what is the "various" models plotted with circles? This is not clear from the caption and should be clarified."*
> >
> > Thank you, we will clarify this in the revision.
> > Various refers to all other models (out of the 523) that were not mentioned by name. You can already see them by name in the plotly plots included in the supplementary.

---

### Decision · Program_Chairs · 2023-01-20

**Decision:**

Accept: poster

**Justification For Why Not Higher Score:**

Due to the nature of the paper, no novel technical insight are presented. Thus, I feel that a poster presentation would be the perfect fit.

**Justification For Why Not Lower Score:**

A rejection would not be suited for the paper since the work is well executed and provides interesting insights.

**Metareview: Summary, Strengths And Weaknesses:**

The authors present a (very thorough!) empirical study on the selective prediction and uncertainty estimation. Being an empirical study, the work does not present a novel technical/algorithmic solution. However, Multiple interesting insights and resulting actions can be deduced from the work. As such, I judge the work as quite interesting for the community. The reviewers mainly agree on these points -- thus, I recommend acceptance.

**Note From Pc:**

if the above contains the word "oral" or "spotlight" please see: "oral" presentation means -> notable-top-5% and "spotlight" means -> notable-top-25%. As stated in our emails, we are disassociating presentation type from AC recommendations